

# A high space-time resolution dataset linking meteorological forcing and hydro-sedimentary response in a mesoscale Mediterranean catchment (Auzon) of the Ardèche region, France

Guillaume Nord[1], Brice Boudevillain[1], Alexis Berne[2], Flora Branger[3], Isabelle Braud[3], Guillaume Dramais[3], Simon Gérard[1], Jérôme Le Coz[3], Cédric Legoût[1], Gilles Molinié[1], Joel Van Baelen[4], Jean-Pierre Vandervaere[1], Julien Andrieu[5], Coralie Aubert[1], Martin Calianno[1,6], Guy Delrieu[1], Jacopo Grazioli[2], Sahar Hachani[1,7], Ivan Horner[3], Jessica Huza[3,8,*], Raphael Le Boursicaud[3], Timothy H. Raupach[2], Adriaan J. Teuling[8], Magdalena Uber[1,9,*], Béatrice Vincendon[10], Annette Wijbrans[1]

[1]IGE, Univ. Grenoble Alpes, CNRS, IRD, F-38000 Grenoble, France
[2]Environmental Remote Sensing Laboratory, École Polytechnique Fédérale de Lausanne, Lausanne, Switzerland
[3]Irstea, UR HHLY, Hydrology-Hydraulics, Villeurbanne, France
[4]LaMP, CNRS/UBP, Clermont Ferrand, France
[5]ESPACE, CNRS/Université - Nice Sophia Antipolis, France
[6]Institut de géographie et durabilité, Université de Lausanne, Lausanne, Switzerland
[7]École Nationale d'Ingénieurs de Tunis, Université de Tunis, El Manar 1, Tunisia
[8]Hydrology and Quantitative Water Management Group, Wageningen University, Wageningen, The Netherlands
[9]Institute of Earth and Environmental Science, University of Potsdam, Potsdam, Germany
[10]GAME/CNRS (Météo-France, CNRS), Toulouse, France
*now at: Amec Foster Wheeler Environment and Infrastructure, Montréal, Canada, 1425 Route Transcanadienne, Dorval, Québec, Canada H9P 2W9

Correspondence to: Guillaume Nord (guillaume.nord@univ-grenoble-alpes.fr)

**Abstract.**

A comprehensive hydrometeorological dataset is presented spanning the period 1 January 2011-31 December 2014 to improve the understanding of the hydrological processes leading to flash floods and the relation between rainfall, runoff, erosion and sediment transport in a mesoscale catchment (Auzon, 116 km²) of the Mediterranean region. Badlands are present in the Auzon catchment and well connected to high gradient channels of bedrock rivers which promotes the transfer of suspended solids downstream. The specificity of the dataset is its high space-time resolution, especially concerning rainfall and the hydrological response which is particularly adapted to the highly spatially variable rainfall events that may occur in this region. This type of dataset is rare in scientific literature because of the quantity and type of sensors for meteorology and surface hydrology. Rainfall data include continuous precipitation measured by rain gauges (5 min time step for the research network of 21 rain gauges and 5min or 1h time step for the operational network of 10 rain gauges), S-band Doppler dual-polarization radars (1 km², 5 min resolution), disdrometers (16 sensors working at 30 s or 1 min time step) and Micro Rain Radars (5 sensors, 100 m height resolution). Additionally, during the special observation period (SOP-1) and enhanced observation period (Sep-Dec 2012, Sep-Dec 2013) of the HyMeX (Hydrological Cycle in the Mediterranean Experiment) project, two X-band radars provided precipitation measurements at very fine spatial and temporal scales (1 ha, 5





min). Meteorological data are taken from the operational surface weather observation stations of Météo-France (including 2-m air temperature, atmospheric pressure, 2-m relative humidity, 10-m wind speed and direction, global radiation) at the hourly time resolution (6 stations in the region of interest). The monitoring of surface hydrology and suspended sediment is multi-scale and based on nested catchments. Three hydrometric stations measure water discharge at a 2 to 10 min time

resolution. Two of these stations also measure additional physico-chemical variables (turbidity, temperature, conductivity) and water samples are collected automatically during floods allowing further geochemical characterization of water and suspended solids. Two experimental plots monitor overland flow and erosion at 1 min time resolution on a hillslope with vineyard. A network of 11 sensors installed in the intermittent hydrographic network continuously measures water level and water temperature in headwater subcatchments (from 0.17 km² to 116 km²) at a time resolution of 2-5 min. A network of soil

moisture sensors enable the continuous measurement of soil volumetric water content at 20 min time resolution at 9 sites. Additionally, opportunistic observations (soil moisture measurements and stream gauging) were performed during floods between 2012 and 2014. The data are appropriate for understanding the rainfall variability in time and space at fine scales, improving areal rainfall estimations and progressing in distributed hydrological and erosion modelling.

**DOI of the referenced dataset:** http://dx.doi.org/10.6096/MISTRALS-HyMeX.1438

## 1 Introduction

There is an increasing need to understand the origin of water and sediment and their transit time within the catchment. According to the pathways and the residence time of water and sediment within the system, the physical, chemical and biological interactions determine the water quality and the nature and composition of sediment and associated matter. Furthermore, to assist the stakeholders in implementing efficient soil and river management measures, the scientific

community aims to understand the processes and the factors that control surface runoff, develop modelling approaches able to provide reliable flow separations, localize sediment sources and sinks, and predict the space-time dynamics of sediment and associated contaminant within the catchment. This requires taking into account the space-time variability of rainfall events, using spatially distributed models and coupling hydrological models with dissolved transport models, erosion and sediment transport models.

Although the benefit of distributed models is recognized for understanding the inner behaviour of the catchment, their reliability do not meet the expectations. Indeed, the water and sediment discharges simulated by distributed models at the outlet of the catchment are generally poorer than the results simulated by lumped models (Jetten et al., 2003; Reed et al., 2004; de Vente el al., 2013). This raises the question of the improvement of distributed models. To date there are various difficulties that hinder the potential of distributed models (e.g. Cea et al., 2016) such as the large number of parameters, the

definition of some parameters which are difficult to measure, the high non-linearity of the equations, the interaction between input parameters, the lack of comprehensive field data available for calibration, the uncertainty in the experimental measurements and input data, and the space-time variability of the physical processes. The preference between event model



and continuous model is also often discussed within the scientific community. The role of antecedent soil moisture conditions on the generation of runoff has been highlighted in the literature (Lelay and Saulnier, 2007; Huza et al., 2014) as it has implication on the estimation of the initial conditions. The deployment of multi-scale observation systems over a period of several years in medium catchments and the release of the collected datasets as open data with metadata on how the data have been collected, quality assured, and their associated uncertainties (Weiler and Beven, 2015) is of crucial importance to exceed the current limitations of distributed models.

The Mediterranean area is prone to intense rainfall events, sometimes triggering flash floods that may have dramatic consequences (Ruin et al., 2008). Flash floods are usually the consequence of short, high-intensity rainfalls mainly of spatially confined convective origin and often enhanced by orography (Borga et al., 2014). As such, flash floods usually impact basins less than 1000 km² (Marchi et al., 2010). In medium-scale Mediterranean catchments, the control exerted by the amount of rainfall and its intensity and variability on the generation of runoff and the erosional processes operating at different scales is of major importance (Navratil et al., 2012; Marra et al., 2014; Tuset et al., 2015). The characterization of rainfall variability is therefore required for hydrological and erosion studies. However the definition of the scales of variability of rainfall and the origins of this variability are still open questions within the hydrometeorological community.

Survey of rainfall all around the world is done mainly using rain gauge networks, ground-based weather radars and in a lesser extent on-board satellites radars or passive radiometers. Usually, the spatial resolution is at best around one kilometre for operational meteorological radars. It can reach several kilometres for precipitation radars on-board satellites and operational rain gauge networks. For the purpose of some specific studies, rainfall has been measured at resolutions ranging from the meter (Anagnostou et al., 1999; Krajwesky et al., 2003) to several hundred of meters (Wood et al., 2000). Other studies (Fraedrich et al., 1993; and Fabry, 1996) have analysed rainfall as a function of the temporal scales, investigating rainfall variability from less than one second to several decades. These studies revealed that rainfall is not steady at scales below few kilometres and minutes, which are typically the scales of the operational rain gauge and radar networks. The deployment of high density rain gauge and disdrometer networks at the mesoscale in combination with research and operational radars is essential. Detailed observations of the rainfall microstructure (size, velocity of hydrometeors) enable to understand the origins of this variability and specifically the role of topography and permit to improve radar rainfall estimations and more generally remote sensing estimations of rainfall amounts, which are highly dependent on the rainfall microstructure.

High space-time resolution datasets linking meteorological forcing and hydro-sedimentary response are rare in scientific literature because of the quantity and the variety of sensors required for measuring rainfall and surface hydrology. The already published datasets consist of first order catchments (Western and Grayson, 1998), catchments where the observation period is exceptionally long (Slaughter et al., 2001; Renard et al., 2008; Stone et al., 2008; Baffaut et al., 2013), or catchments located in snow-dominated mountain (Reba et al., 2011; Kormos et al., 2014). In mesoscale catchments, such datasets are scarce (Goodrich et al., 1997; Gupta et al., 2010), especially in the Mediterranean region.





This study is part of the FloodScale project (Braud et al., 2014), which is a contribution to the HyMeX project (Hydrological Cycle in the Mediterranean Experiment, Drobinski et al., 2014), a 10-year multidisciplinary program on the Mediterranean water cycle. The FloodScale project (2012-2015) encompasses two special observation periods of the HyMeX program, the first one called SOP1 (Ducrocq et al., 2014) was dedicated to heavy precipitation and flash-floods and took place from 5

September to 6 November 2012. The general objective of the FloodScale project is to improve the understanding and simulation of the hydrological processes leading to flash floods. The approach is based on the monitoring of nested spatial scales: (1) the hillslope scale; (2) the small to medium catchment scale (1–100 km²); (3) the larger scale (100–1000 km²). This study focuses on scales that range from the hillslope to the medium catchment scale. Although it is evident that the most extreme events are difficult to observe in small to medium research catchments due to their relatively rare occurrence and

their high destructive potential, the conventional hydrometeorological monitoring networks (rain gauges and stream gauges) may nevertheless provide highly relevant and unprecedented data at high spatial and temporal resolutions allowing documenting small to moderate events. In addition, the duration of the observations (4 years) allows the characterization of the standard catchment behaviour and provides the opportunity to observe less ordinary events with processes that are specific to flash floods and to characterize possible threshold effects that are not observed in small to moderate events. The

observation strategy is reinforced by the deployment of conventional and polarimetric radars that provide precipitation measurements at spatial and temporal scales not properly resolved by rain gauges networks (Berne and Krajewski, 2013). Finally, a special effort is dedicated to soil moisture measurements and stream gauging during floods. These opportunistic observations made possible by a real-time warning system enable to watch transient processes like runoff, to monitor the increase of water content in soil and to gauge high discharges in small to medium catchments, which is an uncertain task due

to the very short response times of such systems. This allows documenting the upper ends of stage-discharge rating curves that are generally extrapolated at high values.

The long-term observation system presented in this study belongs to the Cévennes - Vivarais Mediterranean Hydrometeorological Observatory (OHMCV) (Boudevillain et al., 2011). It is located in Ardèche, in a region with a high gradient in annual rainfall (e.g. Molinié et al., 2012). The observation system has been operated by different teams from

25 various countries during the SOP and EOP HyMeX periods: LTHE, IRSTEA Lyon, EPFL, Wageningen University, LAMP and Météo-France. The data collected during the period from 2011 to 2014 are reported here. The dataset includes precipitation and weather data, soil moisture data, runoff and soil erosion data, hydrologic and suspended sediment response data, surface water quality data, and GIS data.

The paper presents the acquired datasets to make them accessible to the scientific community and make their use easier and

30 wider. The authors are convinced that the published datasets can serve as a benchmark for hydrological distributed modelling applied to the Mediterranean area.



## 2 Catchment description

The Auzon catchment (116 km²) is located in a region traditionally called "Bas Vivarais" between the plains of the Rhône valley to the East (minimum elevation: 40 m a.s.l.) and the Ardèche Mountains to the West (maximum elevation: 1550 m a.s.l.). The Auzon river is a left bank tributary of the Ardèche river which drains from North to South (Fig. 1). The Auzon

catchment ranges in elevation from 140 m a.s.l. to 1019 m a.s.l.. This mid-elevation area includes the volcanic plateau of Coiron to the North (approximately one third of the catchment area), standing as a barrier. The latter closes the horizon over the sedimentary piedmont hills to the South (approximately two thirds of the catchment area). The Coiron plateau is a vast basaltic table ranging in elevation from 600 m to 1000 m that has the appearance of an oak leaf lying on the marly-limestones bedrock of the "Bas Vivarais" (Grillot, 1971; Naud, 1972). An intense volcanic activity between 7.7 and 6.4

million years BP (early Pliocene) produced stacked lava flows and pyroclastic flows that gradually filled a former valley. This phase of volcanic activity is contemporaneous with the volcanism phase of the Mont Mézenc. The current morphology of the region is the result of significant Quaternary erosion which has notched the edges of the lava flows delimiting narrow digitations separated by marly thalwegs. An inverted relief is now observed where the present surface of the plateau corresponds to the former valley bottom. The sedimentary substratum is composed of Cretaceous marls and limestones from

the Upper Jurassic. The former valley bottom is now raised with respect to the young valleys carved by streams and one can observe gorges with steep slopes of marls with typical badlands aspect.

The region is exposed to both oceanic and Mediterranean climatic influences. The terrain of the region is partly mountainous and plays a major role on rainfall properties. The highest average daily rainfall intensities are located over the relief, while the highest average hourly rainfall intensities are located over the plain (Molinié et al., 2012). Average yearly rainfall ranges

between 850 and 900 mm throughout the Auzon basin, which represents an intermediate value between the plains of the Rhône valley to the East (500 mm) and the Ardèche Mountains to the West (2000 mm).

On the Coiron plateau, forest vegetation has almost completely disappeared (Fig. 2 and Fig. 3.b). Oaks, chestnut trees and associated shrub flora only remain on the marly slopes and basalt screes. Mediterranean grazed open woodlands with broom (sarothamnus purgans), boxwood (buxus sempervirens) and sloe tree (Prunus spinosa) cover almost all the rocky outcrops

(Bornand et al., 1977). Grasslands and crops are located in well drained depressions.

On most of limestone formations and marly formations with steep slopes, natural vegetation is dominant. This vegetation consists of downy oak woods (quercus pubescens), garrigues and Mediterranean open woodlands where stunted oaks are associated with broom (sarothamnus purgans), boxwood (buxus sempervirens), juniper and dry grasslands (thyme, aphyllante and Brachypodium). On marly-limestone formations with low slope, the vegetation has been cleared and gave

way to traditional crops (cereals, vines ...) and grazed grassland. Overall, according to the CORINE Land Cover 2006 classification, the Auzon catchment consists of forest (26.7%), pastures under agricultural use (17.1%), vineyards (19%), moors and heathland and sparsely vegetated areas (14.3%), crops (9.5%), natural grasslands (11%), and urban areas (2.3%).



The brown soils on basalt material cover the majority of the volcanic edifice of the Coiron plateau (Fig. 3.c). They gather soils supported by basaltic rock (16a), scoria and tuffs (16b), colluvium of anthropogenic origin (16c), screes and talus fans (16d). These four main soil families constitute a very homogeneous group with very close physico-chemical characteristics. Cartographic differences are based on the nature of the parent rock, the topographic location and the human intervention; those factors determine the depth, the heterogeneity and the texture of soils (Bornand et al., 1977). Soil depths are generally less than 2 m. The soil matrix consists mainly of clay and fine silt. The stony load is variable according to the drainage of the medium.

In the piemont hills beneath the Coiron plateau, there are rendzina (9), clay-stony soils of variable depth (20-70 cm) on marly-limestones and regosols (33a) due to erosion on marls characterized by deep gullies (badlands) that constitute a significant source of suspended material during floods. In less steep terrain, there are generally cultivated soils (13a), loam and clay loam, irregularly deep, decarbonated at the surface, well structured and supporting cereal crops and vines. At the South of the Auzon catchment, there are lithosols and regosols (34a), rocky outcrops and shallow brown calcareous soils (30-40 cm deep) on marly limestones. On the Western edge of the Auzon catchment, rocky outcrops and lithosols (39a and 39b) on Jurassic limestone formations are highly dominant. These karstified formations are responsible for the natural drying up of the Auzon river, frequently observed in its downstream reach. Finally, on the edges of the main rivers (Claduègne and Auzon), calcareous alluvial soils (2b) or coarse textured alluvium (1) are present.

The multi-scale observation system presented in this study (Fig. 3) is based on nested catchments: the Gazel catchment (3.4 km²), the Claduègne catchment (43 km²) and the Auzon catchment (116 km²). Rainfall and weather observations include both operational and research instruments that are located both inside the catchments and in their immediate vicinity. Hydrological observations are mainly concentrated on the Claduègne catchment.

## 3 Data description

Table 1 presents the hydrometeorological variables, gives the characteristics and the number of instruments and indicates whether the measurements belong to an operational network or a research network of observation. Table 2 presents the soil and surface water variables (hydrological and sediment data) and gives the characteristics and the number of instruments. Appendix A is a chart that describes the period of measurement of each instrument (rainfall, meteorology, soil and surface water) and specifies the number of instruments deployed in the field. All the data presented here have undergone careful (mostly manual) quality assurance.



### 3.1 Hydrometeorological data

#### 3.1.1 Rainfall

✓ Radars

The region of interest is covered by two operational S-band radars (Fig. 1): a conventional radar located in Bollène (about 40

5 km away) and a polarimetric radar located in Nîmes (about 90 km away). Their visibility over the Auzon catchment is however hindered by the topography and the lowest beam is at about 2 km above the ground. These operational radar, managed by Météo-France, provided data (radar reflectivity and rain rate estimates) over the entire period of interest. To complement these radars and monitor the small-scale variability of precipitation, two additional X-band research radars were deployed during HyMeX SOP1 (Fig. 3.a), providing measurements at a resolution of about 100x100 m². A radar, managed

by LaMP, provided rapid Plan Position Indicator (PPI) scans (every 3 minutes) at one elevation. EPFL-LTE managed a polarimetric radar that provided a combination of Range Height Indicator (RHI) and PPI scans of polarimetric variables every 5 min. These two research radars enabled the monitoring of low level precipitation over the Auzon catchment. Their maximum range should vary between 30 and 40 km (the range represented in Fig. 3.a is only qualitative). Finally, 5 Micro Rain Radars (MRR), provided by CNRM, LaMP and OSUG, were deployed in combination during Fall 2012 and Fall 2013

at three locations in the region of interest to document the vertical profile of precipitation. These Doppler FW-CW vertical pointing radars measuring the Doppler spectra enable to study the vertical structure of rainfall as well as the associated microphysical processes in relation with the orography (Zwiebel et al., 2015). More detailed information about the operational and research radar systems involved in HyMeX can be found in Bousquet et al (2015). The operational radar processing algorithms are described in Tabary (2007), while the data from the Mobile X-band Polarimetric (MXPol) radar

are processed following the steps described in Schneebeli et al. (2013, 2014). The characteristics of MXPol are summarized in Schneebeli et al (2013).

✓ Disdrometers

A network of 16 OTT Parsivel disdrometers (optical spectropluviometers) including 12 of the first generation and 4 of the second generation covers the Southern part of the Auzon catchment and extends lightly more to the West, up to Saint-

25 Etienne de Fontbellon (Fig. 3.a and Fig. 4). At least 5 devices were available at the same time from 15 November 2011 (see Appendix A for the period of operation of the instruments). Moreover, a 2D Video Disdrometer (2 DVD) was deployed at Le Pradel (south of the Gazel catchment) in Fall 2012 and 2013 for an inter-comparison of measurements (Raupach and Berne, 2015). All Parsivels except Saint-Etienne de Fontbellon and Pradel-Grainage are colocated with rain gauges from the network described in the next paragraph (Fig. 4). The correction technique described by Raupach and Berne (2015) using the

30 2DVD as a reference disdrometer has been applied to Parsivel data, improving the consistency between disdrometers and the collocated rain gauges.



✓ Rain gauges

An operational network of 10 rain gauges (6 managed by Météo-France and 4 managed by the Flood Forecasting Service - SPC Grand Delta) is present over the Auzon catchment or its close vicinity (Fig. 3.a). It provides data at an hourly time step (Météo-France rain gauges) and 5 min time step (SPC Grand Delta rain gauges). Additionally a research network of 21 rain gauges is implemented over the Auzon catchment (Fig. 3.a). It provides data at 5 min time step. 19 rain gauges were initially deployed over a 7x8 km² area located in the Southern part of the catchment (Fig. 4) and 2 additional rain gauges were subsequently installed in the Northern part of the Claduègne catchment. This network called Hpiconet was designed for sampling rainfall at spatial scales ranging from tens of meters to tens of kilometres and at temporal scales ranging from 1 min to 1 day. The rain gauges are gathered on 10 main locations with a mean interdistance of about 2 km. At some locations, several rain gauges are clustered with interdistances of 10 up to 500 m as seen in the inset maps of Figure 4. All rain gauges are identical and consist of Precis Mecanique's tipping bucket of 0.2 mm with a collecting area of 1000 cm². Data collection is made thanks to a Hobo or a Campbell datalogger. A careful data quality control is performed, notably by comparing the total amounts obtained by tipping buckets with those collected at the outlet of the rain gauge with a plastic tank of 30 l. The comparison of these two measurements shows that the relative difference remains below 10% for all the rain gauges and lower than 5% for most of them. Calibration is performed if an error of more than 5% is detected.

### 3.1.2 Other meteorological data

Most of the meteorological variables originate from one of the operational surface observing networks of Météo-France. Six stations are located in the Auzon catchment or its close vicinity. All of them continuously provide hourly measurements. The variables measured at each station differ depending on the network requirements. Figure 3.b shows the location of each station together with the variables they measure.

✓ Air temperature and relative humidity

Each of the six stations considered is equipped with a WMO-standard meteorological shelter, which height is generally 1.5 m. Both an air temperature sensor (PT100) and a relative humidity sensor are mounted in the shelter so that they are protected from solar radiations.

✓ Atmospheric pressure

Atmospheric pressure is measured by a digital barometer (PTB220) at one of the six stations considered.

✓ Wind speed and direction

Wind measurements are conventionally performed at 10 m above ground surface level and on open ground at three of the six stations considered. Wind speed is given by a cup anemometer while wind direction is measured thanks to a vane mounted on a pole that has pointers indicating the principal points of the compass.

✓ Global radiation

Several measurements of radiation can be performed. Two out of the six considered stations are equipped with a pyranometer (CM11) providing global solar radiation values.



Additionally, one Baro-Diver and three Mini-Diver were deployed over the Claduègne catchment in complement of the stream sensors network to measure the atmospheric pressure and the air temperature with an observation frequency of 2 min (Fig. 3.b). However these four sensors are not located in a shelter and are therefore subject to solar radiation.

**3.2 Spatial characterization data**

Characterization data are used to define the topography, pedology, geology, land use, landscape and hydrological properties of the Auzon catchment. These data provide the fine-scale detail required for modelling and hydrological assessment. The coordinate system of reference used in this study is the Réseau géodésique français (RGF) 1993 (official in France, based on IAG-GRS80 ellipsoid, very similar to WGS 84). The projection is Lambert conique conforme. Table 3 presents the main
GIS descriptors available for the region of interest in this study.

**3.2.1 GIS descriptors**

For the Claduègne catchment, a 1 m bare earth digital elevation model (DEM) was derived from an aerial lidar dataset acquired in 2012 and processed by the Sintégra Company (Braud et al., 2014). For the Auzon catchment, the 25 m DEM released by IGN France in 2008 is available. A combination of these two latter DEM was performed using ArcGis based on
re-sampling and interpolation to produce a 5 m DEM over the Auzon catchment from which the catchment boundaries were derived using TAUDEM D8 incorpororated in ArcGis. A map (scale 1:50000) of the geology of the region including the Auzon catchment was released by BRGM in 1996 and digitalized in vector format from 2001 (Elmi et al., 1996). A map (scale 1:100000) of the pedology of the region including the Auzon catchment was released by INRA (Bornand et al., 1977). Besides, the Ardèche soil database at scale 1:250000 produced by Sol-Conseil and Sol Info Rhône Alpes provides a vector
map of the region synthesizing a large amount of information on soil (soil class and unit, horizon, thickness etc) and bedrock. Very high resolution images were acquired and processed to provide detailed land use maps: 5 m resolution satellite images (Quickbird images) taken in 2012 for the Claduègne catchment and 30 m resolution satellite images (Landsat) taken in 2013 for the Ardèche catchment. The ortophotography database released by IGN France in 2009 provides aerial images of the Auzon catchment at a resolution of 0.5 m. In addition, vector data of the drainage network, catchment
boundaries, instrument locations, administrative boundaries and road network are available.

**3.2.2 Infiltration tests**

A field campaign aiming at documenting the variability of surface hydraulic properties was conducted in May–June 2012 in the Claduègne catchment. The measurements were performed at 17 points throughout the catchment (Fig. 5) which were selected from the cross-analysis of pedology, land cover and geology maps following the method of Gonzalez-Sosa et al.
(2010). The tested hypothesis is that land use has a major influence on the observed hydraulic properties rather than the soil texture. Two techniques were used: the mini disk infiltrometer and the double ring infiltrometer using the Beerkan method





(Braud et al., 2005). With the exception of two points, both instruments were used at each location. Between one and three repeated measurements were performed. Soil texture was then analyzed at the INRA laboratory of soil analyses in Arras (France). The results of this campaign are described and released in Braud and Vandervaere (2015).

### 3.3 Hydrological and sediment data

Almost all the instruments deployed in the field to measure the soil and surface water compartments were installed for research purposes. There was virtually no hydrological observation before 2011 in the Auzon catchment, except the water erosion plots and a site for soil moisture measurement (SMOSMANIA network). Most of the instruments were installed in the framework of the FloodScale project and the HyMeX enhanced observation period (2011-2014). The instruments are located mainly in the Gazel and Claduègne sub-catchments (Fig. 3.c) where it was decided to put the major efforts.

### 3.3.1 Surface water

✓   Hydrometric stations

Three hydrometric stations with natural control points are located respectively on the Gazel, Claduègne and Auzon rivers (Fig. 3.c). The Claduègne and Auzon stations are situated at a bridge in order to facilitate the access and the manipulations during floods; they were installed respectively in October 2011 and June 2013. Water level is measured using H-radar (Table

2). The Gazel station is situated on a natural reach without any construction; it was installed in April 2011. Water depth is measured using a hydrostatic pressure probe. The common variables provided by these three stations are water level and stream discharge. The observation frequency is respectively 2 min, 10 min and 5 min for the Gazel, Claduègne and Auzon stations. The logged values are time-averaged measurements (typically 30 values over less than 1 min), with their dispersion (standard deviation, minimum and maximum values). A significant effort was dedicated to the establishment of the stage-

discharge relationships during the period 2012-2014. Many on-alert campaigns were carried out to perform discharge measurements at high flows. All the discharge measurements with their estimated uncertainties at the 95% level of confidence are presented in Table 4. Different techniques including salt dilution, current meter, Surface Velocity Radar (SVR) (Welber et al., 2016), Acoustic Doppler Current Profiler (ADCP), Acoustic Doppler Velocimeter (ADV) and Large Scale Particle Image Velocimetry (LSPIV), based on images recorded by a fixed camera (Le Coz et al., 2010; Dramais et al.,

2015) were used depending on the type of river and the hydraulic conditions. The BaRatin framework (Le Coz et al., 2014) combining Bayesian inference and hydraulic analysis was used to build steady, multi-segment stage-discharge relationships and to estimate the associated uncertainty (95% confidence interval).

Additional variables are provided by these stations. At the Gazel and Claduègne stations, different physico-chemical variables of the surface water are measured continuously: temperature, conductivity and turbidity. Sequential samplers,

triggered by the data logger, collect water and suspended sediment samples when threshold values of water level and turbidity are exceeded. Water samples are oven dried and weighed in the lab to calculate Suspended Sediment Concentration (SSC). Some selected samples are analysed using a laser diffraction particle size analyser (Malvern Mastersizer/E) to



characterize the particle size distribution. At the Claduègne station, water surface velocity is measured continuously using the non-contact radar technology based on the principle of the frequency shift due to the Doppler Effect. The continuous measurement of water velocity has become increasingly common in the US and in Europe, especially for operational hydrometric agencies, as it allows applying the Index Velocity Method (Levesque et al., 2012). This approach is particularly

relevant to small rivers subject to flash floods where flow is highly unsteady. It represents a useful tool for extrapolating stage-discharge rating curves over a range of flows for which the use of conventional gauging methods is impractical or unsafe (Nord et al., 2014). At the Claduègne station, an acoustic Doppler velocity meter was fixed to the channel bed during the period from September 2013 to November 2014 to measure detailed velocity profile (100 cell maximum) at the same observation frequency as water level and surface velocity. This system provides an alternative continuous measurement of

flow velocity in the water column from the bed up to the water surface.

✓    Overland flow and water erosion

Two erosion plots were monitored on a hillslope with vineyard at "Le Pradel" (Fig. 2.e and Fig. 4) during the period from December 2009 to October 2013. The erosion plots, considered as two duplicates, are 60 m long and 2.2 m width and they extend over the entire length of the hillslope. The width of the plots corresponds to the distance between two vine rows

oriented in the direction of the main slope. The vine rows are located on the edges of each plot. The two plots are parallel and spaced by approximately five meters. The average slope in the longitudinal direction is about 15%. The vegetation cover between the vine rows varied between years but remained very sparse. The brown calcareous soils underlain by marly-limestones are composed with 34% of clay, 41% of silt and 25% of sand particles. The Gazel river is located about 40 meters away from the monitored hill foot. The transition between the cultivated hillslope and the river is marked by a riparian

vegetation zone and a cliff of about 10 m. This monitored hillslope is included in the catchment whose outlet corresponds to the Gazel hydrometric station with the idea of investigating the fate of solid particles eroded from the hillslope to the river.

A rain gauge and a disdrometer were located at about 30 m from the erosion plots (Fig. 4). The two plots were equipped similarly. Runoff was collected in the bottom part of the hillslope. The water depth was measured every minute with a 1 mm resolution using a gauge (OTT Thalimede) within a H-flume designed following the US Soil Conservation Service

recommendations. The stage-discharge rating curve was built experimentally and allowed to calculate discharge with a median relative uncertainty of 10%. A sequential sampler containing 24 bottles of 1 l capacity sampled water and eroded particles within the H-flume. When critical thresholds of water depth or water depth variation were exceeded, the data logger triggered the sampling of water and eroded particles. Thus, the time intervals between each two samples were irregular, depending on the shape of the hydrograph. The suspended sediment concentrations were estimated by weighting the water

samples after drying them during 24 h at 105 °C with a median relative uncertainty of 15%. While the discharges were available continuously, the sediment fluxes were only calculated for the times where suspended sediment concentrations were available. Many samples were analysed using a laser diffraction particle size analyser (Malvern Mastersizer/E) to characterize the particle size distribution. More details about the description of the plots, the topographical data available,



and the monitored runoff and erosion events are given by Grangeon (2012) and Cea et al. (2016). The infiltration and runoff processes over this hillslope were previously studied by Nicolas (2010).

✓ Stream sensors network

A stream sensors network composed of four CTD-Divers (Conductivity Temperature Depth) and seven Mini-Divers
(Temperature Depth) was deployed on the Gazel and Claduègne catchments (Fig. 3.c). These compact instruments (Table 2) for autonomous measurement and record were installed in small metallic boxes (177 x 81 x 57 mm) embedded in the riverbed in the case of bedrock rivers and anchored vertically to the wall or any other fixed element in the rest of cases. Anyway, two rules prevailed in the setting up of the sensors: 1) they should be located as close as possible to the riverbed to enable the longest possible continuous recording, including during periods with very low flows; 2) they must be protected
against block impacts related to bed load transport. The lids of the boxes were perforated to ensure water permeability. In addition, in the case of the CTD-Divers, a hole of 3 cm in diameter were formed at each end of the boxes to let circulate the water and ensure a significant renewal of water inside. The instruments were installed in the intermittent hydrographic network, delineating ten sub-catchments of 0.17 to 2.2 km² and one sub-catchment of 12.2 km² which contains the whole area with volcanic geology of the upstream part of the Claduègne catchment. The selected sites are mainly headwater sub-
catchments where the landscape properties are considered homogeneous in terms of geology, pedology and land use (Fig. 3). The underlying assumption in the choice of the measurement sites was that the delineated sub-catchments were homogeneous hydrological units and could lead to different responses for the same rainfall forcing. Taken together, these selected sub-catchments constituted a representative sample of the landscapes encountered in the Gazel and Claduègne catchments. Given that rainfall is highly variable in space and time in this region, the observation system enables to estimate
rainfall fields with a spatio-temporal resolution (typically 1 km² and 15 min) adapted to the size of the delineated sub-cachtments of the stream sensors network.

The CTD-Divers and Mini-Divers measure the total pressure as they are not compensated (cableless instruments). An independent measurement of atmospheric pressure is therefore necessary for accurate barometric compensation and consecutive calculation of the hydrostatic pressure or water depth. Initially (in September 2012), only one barometer (Baro-
25 Diver) was installed in the area following the manufacturer's recommendation which specifies that, in general, in relatively flat open terrain, the pressure measurement has a maximum range of 15 km. However, the error in the measurement of water level was important in our case (about 2 cm), mainly due to the error in the atmospheric pressure which varies significantly throughout the area due to the relief and the differences of climate between the Coiron plateau and the piedmont hills. As a consequence 3 additional Mini-Divers (used as barometers) were progressively deployed from November 2012 to April 2013
in the Gazel and Claduègne catchments to reduce the measurement error of water level to about 1 cm. In order to compensate the total pressure values measured by the CTD-Divers and Mini-Divers, it is necessary to calculate the atmospheric pressure at all points of the stream sensors network. For this, we rely on the 4 points of atmospheric pressure measurement available and we choose between the following two options according to criteria of distance and difference of altitude between the calculated point and the measuring points:



1) linear interpolation of atmospheric pressure between the two closest points of measurement based on the difference of altitude with the calculated point

2) meteorological method of correcting pressure (National Advisory Committee for Aeronautics, 1954) based on the nearest point of pressure and temperature measurement by applying a standard temperature gradient (-6.5 K km$^{-1}$).

The results are not very sensitive to the method used, the most sensitive factor being the density of atmospheric pressure measurement over the spatial extension of the stream sensors network.

When possible, controlled sections are chosen to allow the establishment of a stage–discharge relationship based on stability and sensitivity of the control points. This is the case for three points of the stream sensors networks: mi3, sj1, and vb1 (Fig. 3.c). Mi3 is located on a concrete, broad-crested artificial control, sj1 on a natural weir and vb1 in a circular concrete culvert.

Many on-alert campaigns were carried out to perform discharge measurements at different flow conditions at these points and at two additional points (bz1 and sg1) for which stage-discharge relationships could be established in the future.

### 3.3.2 Soil

✓ Soil moisture

Infiltration excess runoff was thought to be the dominant process (e.g. Nicolas, 2010) in the Gazel and Claduègne

catchments. The observation strategy thus focuses on the documentation of the soil infiltration capacity and initiation of ponded conditions at the surface. The monitoring of soil moisture in the Gazel and Claduègne catchments is a task that has two components:

1. Mobile (manual) soil moisture measurements at the surface before/after rainfall events (good accuracy on the average);

2. Deployment and maintenance of fixed sensors (continuous monitoring but point values).

A series of mobile soil moisture measurements were conducted in the Gazel catchment during the HyMeX SOP1 (Huza et al., 2014). The measurements were taken on 6 fields, distributed on the whole catchment (Fig. 5). Fields were selected to appropriately represent the catchment, while still capturing inter-field variability and the influence of different topographical features. Vineyards were not selected because the soil was dominated by stones, making it impossible to sample without breaking the sensor. This resulted in all selected fields being pastures and grasslands. Within each of the selected six fields, a

transect path of 50 m was measured. Along the 50 m transects, a measurement was taken at spatial intervals of 2 m and all measurements were done at the same location for each of the measurement days. Point volumetric soil moisture measurements were done using a portable three-prong (6 cm rod length) ThetaProbe unit (Delta-T Devices Ltd, Cambridge, UK), which employs the time domain reflectometry (TDR) technique. On each measurement day, all fields were measured within a few hours to minimize the influence of evaporation and drainage processes. The strategy was to select

measurements days that aligned with high precipitation events and to capture both pre-event and post-event soil moisture conditions whenever possible. During SOP1, 16 measurement days were completed on the six different transects. This produced approximately 2500 soil moisture measurements. The accuracy is ± 0.01 cm$^3$ cm$^{-3}$.



Nine sites (Fig. 3 and Fig. 5) with different land uses (two vineyards, four pastures, one piece of fallow land, two small oak woods) were selected for the installation of fixed sensors. Six sites are located in the Gazel catchment (2 in the mi3 and 1 in the mi4 sub-catchments) or in its close vicinity and are representative of the piedmont hills landscapes (Fig. 2.d and e). Three sites are located in the bz1 sub-catchment or its close vicinity and are representative of the Coiron plateau landscapes (Fig.

5    2.a). These choices of localisation were motivated by the presence of the stream sensors network with the objective to make the most direct connection possible between rainfall forcing and hydrological response in small catchments relatively homogeneous in terms of geology and land use. The nine sites were equipped in 2013 with 5 sensors for continuous soil moisture measurements: two at about 10 cm, two at 20–25 cm and one at 30–50 cm depth, in order to document soil saturation (Braud et al., 2014). These 5 sensors are connected to the same datalogger and the observation frequency is 20

10    min (Nicoud, 2015; Uber, 2016). The selected sensors are capacitive probes (Decagon 10 HS) which measure the dielectric constant of the soil in order to find its volumetric water content. The sensors consist of an energy source and two plane electrodes (145 mm long). The soil and electrodes form a capacitor. The dynamics of electricity travelling into the soil depends on the water and dissolved salts content. The sensors come pre-calibrated. Soil water content measurement ranges between 0 and 0.57 $cm^3$ $cm^{-3}$. The accuracy is ± 0.03 $cm^3$ $cm^{-3}$.

Additionally, a station of SMOSMANIA (Soil Moisture Observing System - Meteorological Automatic Network Integrated Application) is located in the close vicinity of the upstream part of the Claduègne catchment, on the Coiron plateau (Fig. 3.c and Fig. 5). SMOSMANIA is based on the existing network of operational weather stations of Météo-France (RADOME). Among the 21 stations of this network that compose an Atlantic-Mediterranean transect in the Southern part of France, Berzème is the station of interest for this study. The land cover around the station consists of fallow, cut once or twice a year.

Four probes measuring soil moisture (ThetaProbe ML2X) were installed at the following depths: 5, 10, 20 and 30 cm.

  ✓  Soil temperature

Soil temperature is measured at the station of SMOSMANIA (Berzème) at the following depths: 5, 10, 20 and 30 cm. The accuracy is ± 0.5 °C.

## 4 Example of data use

### 4.1 Areal rainfall estimation

Areal rainfall estimations are important for water budget assessment and the understanding of the internal catchment behaviour. Geostatistical techniques (ordinary kriging and kriging with external drift) were used to obtain radar–rain gauge quantitative precipitation estimates (QPE's). The uncertainty of these QPE's was calculated using the methodology presented by Delrieu et al. (2014). The QPE's were produced for a wide range of spatial and temporal resolutions (15 to 360

30    min, 1 to 300 km²) for a 30 by 32 km² window encompassing the Auzon catchment, in order to assess the effect of adding high resolution rainfall data on the quality of the QPE for small scale hydrology applications (Wijbrans et al., 2014). Rainfall estimates and error structure were compared for 4 scenarios with varying rainfall datasets (operational rain gauges,





operational + research rain gauges, operational rain gauges + radar, all data) for the 25 largest rainfall events of 2012 and 2013. The results show that the error of the QPE is larger for smaller spatio-temporal resolutions, and that the largest differences between the scenarios are for smaller resolutions as well. The added value of dense rainfall data for the larger spatio-temporal resolutions is limited. The decrease in QPE uncertainty when adding the research rain gauge network is

similar to the decrease when adding the radar data, however the spatial structure of the errors and the rainfall estimates of these scenarios show large differences.

Additionally, a significant effort has been dedicated to the production of rainfall re-analysis for the 2007-2014 period (Boudevillain et al., 2016) based on the operational radar and rain gauge data for a window of 32000 km² including the major catchments of the Cévennes region (Doux, Eyrieux, Cance, Ardèche, Cèze, Gardons, Vidourle, Hérault). These QPE's

were produced with at daily and hourly time resolution and for two types of geographic discretisation: 1) Cartesian meshes of 1 km² for a regular grid covering the study area; 2) "hydrological" units corresponding to the discretisation of the major catchments in subcatchments of homogeneous size in the range of 5, 10, 50, 100, 200, 300 km². The uncertainty of these QPE's was also calculated using the methodology presented by Delrieu et al. (2014). An example of these QPE's is shown in Fig. 6 for the region of the Auzon catchment during the 04 November 2014 event between 13:00 and 14:00 UTC which

corresponds to the peak of rainfall preceding the peak of discharge measured at the Auzon station (Fig. 8). The data presented in this section are made available in the released dataset associated to this paper.

**4.2 Improvement of the quantification of flood hydrographs and reduction of their uncertainty**

The Gazel-Claduègne-Auzon experimental data were also used to develop a methodology to quantify and reduce the uncertainty of flood hydrographs. This methodology is based on the non-contact stream gaugings performed during the on-

alert campaigns (see 3.3.1. "Hydrometric stations") and on the BaRatin framework (Le Coz et al., 2014). On the Auzon hydrometric station, during the 2014 campaign, 11 LSPIV gaugings could be performed through the automated station. They were completed by 10 SVR gaugings. These gaugings have higher uncertainties than the traditional dilution or velocity-area methods, but have the advantage of being feasible safely even under hazardous, high flow conditions. These gaugings were incorporated as observations in the BaRatin methodology, which was further developed by adding the propagation of stage

uncertainty and rating curve uncertainty to discharge time series (Horner, 2014; Branger et al., 2015). BaRatin is based on hydraulic analysis of the flow conditions at the stations, which are used as priors. The Bayesian framework then calculates the posterior rating curve and its associated uncertainty by incorporating the uncertain gaugings. Figure 7 shows that for the Auzon station, the new gaugings contributed to establish a rating curve significantly different from the prior rating curve, and different from the one which could have been obtained using traditional gauging methods only. The difference is

particularly important for high flow. The rating curve uncertainty is also significantly reduced. As far as discharge estimates are concerned, we saw during the events of the 2014 campaign that peak discharge was 25% to 30% underestimated, and that uncertainty on the peak value could be reduced from 80% to 12% only (Fig. 8). The estimation of the flow volume during the event is also 24% underestimated without the noncontact gaugings. This improved accuracy in peak discharge and flow





volume estimation is precious for the validation of hydrological models, but also for more applied purposes (flood forecasting, mapping of flood risk areas), along with estimation of rainfall uncertainty. In the near future, this methodology will be further extended to other hydrological indicators derived from discharge time series, including water balance and drought.

## 4.3 Distributed physically-based soil erosion modelling

The impact of model simplifications on soil erosion predictions was tested applying the GLUE methodology to a distributed event-based model at the hillslope scale (Cea et al., 2016). In this paper the authors analysed how the performance and calibration of a distributed event-based soil erosion model at the hillslope scale is affected by different simplifications on the parameterisations used to compute the production of suspended sediment by rainfall and runoff. Six modelling scenarios of different complexity were used to evaluate the temporal variability of the sedimentograph at the outlet of a 60 m long cultivated hillslope. The six scenarios were calibrated within the GLUE framework in order to account for parameter uncertainty, and their performance was evaluated against experimental data registered during five storm events. The NSE, PBIAS and coverage performance ratios showed that the sedimentary response of the hillslope in terms of mass flux of eroded soil can be efficiently captured by a model structure including only two soil erodibility parameters which control the rainfall and runoff production of suspended sediment. Increasing the number of parameters makes the calibration process more complex without increasing in a noticeable manner the predictive capability of the model.

## 5 Data availability

As an example of the kind of data made available in this paper, Figure 9 shows an overview of rainfall, discharge and turbidity for the entire record (period 2011-2014) at the Claduègne hydrometric station. The measurement period is characterized by a wide variety of water conditions: a dry year in 2012, a wet year in 2014 and an intermediate year in 2013, some intense rainfall events in spring 2013, fall-winter 2013-2014 and fall 2014. Note that the 4 November 2014 flood is a 5-10 years return period flood for the Claduègne river.

All the individual datasets presented in this study are listed in Table 5 and Table 6. They correspond to data collected by a specific instrument, a network of instruments or a type of GIS descriptors. This granularity enables to associate each dataset with a Principal Investigator that is very familiar with the data and who will be an essential resource for any user in case of need. In addition, the added-value dataset corresponding to the results of the rainfall re-analysis for the 2007-2014 period (Boudevillain et al., 2016) based on the operational radar and rain gauge data has been added to Table 5. All the individual datasets are available through the Hymex Database (http://mistrals.sedoo.fr/HyMeX/) maintained by the ESPRI/IPSL and SEDOO/Observatoire Midi-Pyrénées in France as shown in Table 5 and Table 6 which present a list of url links. For each dataset, the metadata include the dataset name, the period of observation, the principal investigator in charge of the dataset, a description of the data, the geographic coordinates of the instruments, a description of the instruments and the measured





variables, the format of the data and the data policy. Most of the individual datasets (32 of 41) have "public" access. A few of them (9 of 41) are subject to the registration step of the Hymex Database as "Associated scientists" (http://mistrals.sedoo.fr/HyMeX/Data-Policy/HyMeX_DataPolicy.pdf). Additionally a bundling service was performed to facilitate the use of the data. The bundled data include the most commonly used data in hydrometeorological and

hydrological studies. The bundled data presents the advantage of gathering data in ASCII and cartesian format, and in a single coordinate system. The bundled data are selected for the spatial and temporal windows presented in the paper since some individual datasets have different extents. 12 individual datasets out of 41 presented in the paper are not part of these bundled data since the effort to prepare the data was too heavy and their potential use is more restricted. These datasets remain accessible individually even though they are not necessarily in the same format and with the same extent (Polar

versus Cartesian and coordinate system). The bundled data are organized in two zip files: the "zip1" file is for the data with "public" access while the "zip2" file is for the data subject to the registration step of the Hymex database as "Associated scientists".

Some of the datasets are also stored on the OHMCV website (http://www.ohmcv.fr), and accessible via the SEVnOL system, maintained by LTHE. Part of the stage and flow time series are also available through the public BDOH database

(https://bdoh.irstea.fr/OHM-CV/) maintained by IRSTEA and managed by the data producers (LTHE, IRSTEA). SEVnOL and BDOH are complementary tools to the bundling service proposed in this study (through the release of the zip files). BDOH was developed for the management of long-term time series and enables the following features: visualization of data, downloading of data, interpolation of time steps for export, import and export to multiple formats, automatic calculation of derived time series. SEVnOL is a Web interface developed to view and extract data, metadata and products in several

formats (XML, CSV, NetCDF) over a user-defined spatial and temporal window (Boudevillain et al., 2011).

**6 Conclusion**

A high space-time resolution dataset linking hydrometeorological forcing and hydro-sedimentary response in a mesoscale catchment is presented. The Auzon catchment (116 km²), a tributary of the Ardèche river, is subject to precipitating systems of Mediterranean origin which can result in significant rainfall amount. The data presented cover a period of four years

(2011-2014) including the HyMeX-SOP1 field campaign (Ducrocq et al., 2014) and the ANR FloodScale project (Braud et al., 2014) which aims at improving the understanding of processes triggering flash floods. The multi-scale observation system presented is part of the OHMCV (Boudevillain et al., 2011). The precipitation measurement is extensive, both in quantity (intensity, volume) and quality (size, fall velocity of hydrometeors). The operational and research networks provide high space-time resolution data (<1 km², 5 min) for studying the microphysics of precipitating systems and producing QPE

particularly adapted to fine-scale hydrological studies. The measurement of the other meteorological variables relies almost exclusively on the operational network (1h time resolution). Validation data are both spatially distributed and multi-scale. They include point measurements of soil moisture (fixed sensors in continuous mode and mobile sensors during rain events),



runoff and erosion measurements on hillslope, water level measurements in the intermittent hydrographic network of headwater catchments (11 points of measurement) and hydrometric measurements (discharge, water conductivity and temperature) at the outlet of 3 nested catchments (3.4, 44 and 116 km²). Discharge measurements were made at high water levels during on-alert campaigns to establish stage-discharge relationships. It is hoped that using this dataset will lead to

advances in understanding hydrological processes leading to flash floods and improving distributed hydrological models.

*Author contribution*

G. Nord, B. Boudevillain, A. Berne, G. Dramais, C. Legout, G. Molinié, J. Van Baelen, and J.P. Vandervaere were Principal Investigators responsible for specific instruments or networks of instruments which resulted in the main individual datasets presented in this study. I. Braud was the leader of the FloodScale (2012-2015) ANR project. F. Branger, I. Braud, J. Le Coz,

G. Delrieu, G. Nord, and J.P. Vandervare were responsible of Work Packages within the FloodScale project which contributed significantly to the design of the observation system presented in this study. S. Gérard, M. Calianno, and C. Aubert helped in installing and maintaining the observation system. J. Andrieu prepared the land use maps. G. Nord prepared the manuscript with contributions from all co-authors. F. Branger and I. Horner helped elaborating Figure 7 and 8. G. Nord, B. Boudevillain, and I. Braud managed the process of data preparation and DOI attribution to the individual datasets for

subsequent submission to ESSD journal. G. Nord and B. Boudevillain contributed to the bundling of the data. B. Boudevillain performed the selection of the individual datasets for the spatial and temporal windows presented in the paper. All co-authors contributed to the data collection or criticizing of the individual datasets presented in this study.

*Acknowledgements*

The FloodScale project is funded by the French National Research Agency (ANR) under contract no. ANR 2011 BS56 027,

which contributes to the HyMeX program. It also benefits from funding by the MISTRALS/HyMeX program (http://www.mistrals-home.org). OHMCV is supported by the Institut National des Sciences de l'Univers (INSU/CNRS), the French Ministry for Education and Research, the Environment Research Cluster of the Rhône-Alpes Region, the Observatoire des Sciences de l'Univers de Grenoble (OSUG/Grenoble University) and the SOERE Réseau des Bassins Versants (Alliance Allenvi). The development of the BDOH database was supported by Irstea internal funding and the

Rhône Sediment Observatory (OSR) project, partly funded by the Plan Rhône. The equipment of the erosion plots and part of the Gazel station was funded by Rhône-Alpes Region. The Sontek-IQ Plus instrument was funded by the French National Research Agency (ANR) under contract no. ANR-12-JS06-0006 (SCAF project). The authors thank the providers of operational data: Météo-France and the SPC Grand Delta. The contract of S. Gerard was funded by the Institut National des Sciences de l'Univers (INSU/CNRS). The Ph.D. thesis of A. Wijbrans was funded by Region Rhône-Alpes. We warmly

thank colleagues of LTHE (especially Romain Biron who participated at the beginning of the project), including students and



technical staff (with special thoughts to Matthieu Le Gall), who helped in installing and maintaining the observation system. We also thank Isabella Zin and Jeremy Chardon for providing the analog rainfall forecasting. LTHE is part of Labex OSUG@2020 (ANR10 LABX56) which funded the contract of Martin Calianno who also participated to the first installations. The HyMeX database teams (ESPRI/IPSL and SEDOO/Observatoire Midi-Pyrénées) and the team of the OSUG Data Center helped in accessing the data and attributing DOIs to the individual datasets. In addition, the authors acknowledge the EPLEFPA Olivier de Serres, the municipalities of Lavilledieu, Lussas, Mirabel, Saint-Germain, Saint-Etienne de Fontbellon, and Villeneuve de Berg, and the local landowners and neighbours for hosting the experiments. T. Raupach acknowledges the support from the Swiss National Science Foundation.

*Acronyms*

LTHE: Laboratoire d'Etude des Transferts en Hydrologie et Environnement, Laboratory for the study of Transfers in Hydrology and Environment

IRSTEA: Institut national de Recherche en Sciences et Technologies pour l'Environnement et l'Agriculture, National Research Institute of Science and Technology for Environment and Agriculture

EPFL: École Polytechnique Fédérale de Lausanne, Swiss Federal Institute of Technology

LAMP : Laboratoire de Météorologie Physique, Laboratory of Physical Meteorology

ESPRI/IPSL: Ensemble de Services Pour la Recherche de l'Institut Pierre Simon Laplace, Research oriented services of the Pierre Simon Laplace Institute.

SEDOO: Service de DOnnées de l'Observatoire Midi-Pyrénées, Data management service of the Midi-Pyrénées Observatory.

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



## Tables and Figures

**Table 1: Overview of the instruments used to gather the hydrometeorological variables in the region that encompasses the Auzon catchment (Ardèche, France), between 2011 and 2014. Note that RHI means « Range Height Indicator », PPI means « Plan Position Indicator », Op means « Operational », and Res means « Research ».**

| Compartment | Op / Res | Instrument | Variable | Unit | Number | Observation frequency | Integration method |
|---|---|---|---|---|---|---|---|
| Rainfall | Op | S-BAND Doppler and Polarimetric radar | reflectivity | dBZ | 2 | 5 min | instantaneous |
| | | | cumulative rainfall | mm | 2 | | |
| | Res | X-BAND Doppler and Polarimetric radar MXPol | horizontal reflectivity | dBZ | 1 | 5 min | instantaneous |
| | | | differential reflectivity | dBZ | 1 | Combination of RHI and PPI scans | |
| | | | Differential phase | ° | 1 | | |
| | | | Doppler power spectra | dBm | 1 | | |
| | | | cross-spectra | dBm | 1 | | |
| | Res | X-BAND fast-scanning radar WR-10X+ | reflectivity | dBZ | 1 | 3 min | instantaneous |
| | Res | micro rain radar MRR-2 | reflectivity | dBZ | 3 | 10 s resolution - 1 min average | integrated |
| | Res | Disdrometer Parsivel 1 | drop size distribution | mm/m$^3$ | 12 | 10 s resolution - 1 min average | integrated |
| | | | drop velocity distribution | | 12 | | |
| | | | precipitation rate | mm/h | 12 | | |
| | Res | Disdrometer Parsivel 2 | drop size distribution | mm/m$^3$ | 4 | 10 s resolution - 30 s or 1 min average | integrated |
| | | | drop velocity distribution | | 4 | | |
| | | | precipitation rate | mm/h | 4 | | |
| | Op | rain gauge Météo-France, SPC Grand Delta | cumulative rainfall | mm | 10 | 1 h (Météo-France), 5 min (SPC) | integrated |
| | Res | rain gauge Hpiconet | cumulative rainfall | mm | 21 | 5 min | integrated |
| Meteorology | Op | Temperature probe PT100 | air temperature | °C | 6 | 1 h | instantaneous |
| | Res | Baro-Diver DI500 , Mini-Diver DI501 | air temperature | °C | 4 | 2 min | instantaneous |
| | | | atmospheric pressure | cm H$_2$O | 4 | 2 min | instantaneous |
| | Op | Humidity probe HMP45D | relative humidity | %RH | 6 | 1 h | instantaneous |
| | Op | Barometer PTB220 | atmospheric pressure | hPa | 1 | 1 h | instantaneous |
| | Op | Wind sensor DEOLIA 96, Alizia 312 | wind speed | m/s | 3 | 1 h | instantaneous |
| | | | wind direction | ° | 3 | 1 h | instantaneous |
| | Op | Pyranometer Kipp & Zonen | inc. shortwave/longwave radiation | W/m$^2$ | 2 | 1 h | instantaneous |

\* The column "Number" indicates the maximum number of instruments in operation at the same time.



**Table 2: Overview of the instruments used to gather the hydrological and suspended sediment variables in the Auzon catchment (Ardèche, France) or its close vicinity, between 2011 and 2014.**

| Compartment | Instrumental device | Instrument | Variable | Unit | Number | Observation frequency |
|---|---|---|---|---|---|---|
| Surface water | Hydrometric stations | Pressure Probe PLS | water level | m | 1 point | 2 min |
| | | | discharge | m | 1 point | 2 min |
| | | Radar level sensor Cruzoe | water level | m | 1 point | 10 min |
| | | | discharge | m | 1 point | 10 min |
| | | Radar level sensor RLS | water level | m | 1 point | 5 min or 1 h |
| | | | discharge | m | 1 point | 5 min or 1 h |
| | | LSPIV / analogical camera VW-BP330 | water surface velocity | m/s | 1 point | 5 min or 1 h |
| | | Radar surface velocity sensor RG-30 | water surface velocity | m/s | 1 point | 10 min |
| | | Acoustic Doppler Velocimeter IQ Plus | water velocity profile | m/s | 1 point | 10 min |
| | | Conductivity and Temp. Probe CS547 | water conductivity | µS/cm | 2 points | 2 min or 10 min |
| | | | water temperature | °C | 2 points | 2 min or 10 min |
| | | Suspended Solids probe Visolid IQ 700 | turbidity | g/l SiO2 | 2 points | 2 min or 10 min |
| | | 3700 Portable sampler | sediment concentration | g/l | 2 points | 10 min or 40 min |
| | Water erosion plots | HS Flume with level sensor Thalimedes | discharge | l/s | 2 points | 1 min |
| | | 3700 Portable sampler | sediment concentration | g/l | 2 points | variable |
| | Stream sensors network | Mini-Diver DI501 | water level | m | 7 points | 2 min |
| | | | water temperature | °C | 7 points | 2 min |
| | | CTD-Diver DI271 | water level | m | 4 points | 2 min or 5 min |
| | | | water temperature | °C | 4 points | 2 min or 5 min |
| | | | water conductivity | µS/cm | 4 points | 2 min or 5 min |
| | | | discharge | l/s | 3 points | 2 min or 5 min |
| Soil | Soil moisture network | Theta Probe | soil volumetric water content | $m^3/m^3$ | 6 transects x 25 points | pre- and post-event |
| | | 10 HS | soil volumetric water content | $m^3/m^3$ | 9 profiles with 5 sensors | 20 min |
| | | ThetaProbe ML2X | soil volumetric water content | $m^3/m^3$ | 1 profile with 4 sensors | 1 h |
| | | | soil temperature | °C | 1 profile with 4 sensors | 1 h |




**Table 3: List of GIS descriptors available for the Auzon catchment.**

| GIS descriptor | Data | Type | Date | Legend | Author | Access |
|---|---|---|---|---|---|---|
| Topography | 1 m bare earth DEM of Claduègne catchment | raster | 2012 | LiDAR campaign | Sintégra géomètres experts | public |
| | 25 m bare earth DEM of Auzon catchment | raster | 2008 | BD TOPO, Ardèche | IGN France | subject to licensing terms |
| | 5 m bare earth DEM of Auzon catchment | raster | 2014 | Combination of data based on resampling and interpolation | LTHE | public |
| Geology | map of Auzon catchment at scale 1:50000 | vector | 1996 | BD Charm-50, Aubenas | BRGM | subject to licensing terms |
| Pedology | map of Auzon catchment at scale 1:100000 | raster | 1977 | Pedological map of France at 1:100000, Privas | INRA | subject to licensing terms |
| | map of Auzon catchment at scale 1:250000 | vector | 2001 | IGCS - Référentiel Régional Pédologique, BDSol-Ardèche | BRGM/Chambre Agriculture | subject to licensing terms |
| Soil properties | soil depth for each SCU* | vector | 2015 | Processed from the Ardèche soil database at 1:100000 | IRSTEA Lyon | public |
| | maximum soil water storage for each SCU* | vector | 2015 | Processed from the Ardèche soil database at 1:100000 | IRSTEA Lyon | public |
| | soil texture of superficial layer for each SCU* | vector | 2015 | Processed from the Ardèche soil database at 1:100000 | IRSTEA Lyon | public |
| | soil stone content for each SCU* | vector | 2015 | Processed from the Ardèche soil database at 1:100000 | IRSTEA Lyon | public |
| Infiltration tests | infiltration campaign Claduègne catchment | vector | 2012 | 52 sampled points | IRSTEA Lyon | public |
| Land use | 5 m resolution images of Claduègne catchment | vector/raster | 2012 | Processed from Quickbird images | UMR Espace | public |
| | 30 m resolution images of Auzon catchment | raster | 2013 | Processed from Landsat images | UMR Espace | public |
| Orthophotography | 0.5 m resolution images of Auzon catchment | raster | 2009 | BD ORTHO, Ardèche | IGN France | subject to licensing terms |
| Surface information | catchment boundaries | vector | 2014 | Processed from the 5 m bare earth DEM with TAUDEM D8 tool | LTHE | public |
| | drainage network (stream) | vector | 2010 | BD CARTHAGE | Sandre eaufrance | public |
| | drainage network (permanent and intermittent) | vector | 2008 | BD TOPO, Ardèche | IGN France | subject to licensing terms |
| | instruments | vector | 2014 | Point | LTHE | public |
| | road network | vector | 2008 | BD TOPO, Ardèche | IGN France | subject to licensing terms |
| | admnistrative boundaries | vector | 2008 | BD TOPO, Ardèche | IGN France | subject to licensing terms |

* SCU means Soil Cartographic Unit.





**Table 4: List of the discharge measurements carried out at the three hydrometric stations (Gazel, Claduègne, Auzon) between 2011 and 2014. The gauging techniques include salt dilution, current meter, Surface Velocity Radar (SVR), Acoustic Doppler Current Profiler (ADCP), Acoustic Doppler Velocimer (ADV), Large Scale Particle Image Velocimetry (LSPIV), and Manning-Strickler.**

| hydrometric station | date | stage | discharge | expanded uncertainty (95% confidence level) | method |
|---|---|---|---|---|---|
| | (TU) | (cm) | (L/s) | (%) | |
| Gazel | 08/11/2011 11:03 | 11.5 | 240.4 | 10 | salt dilution |
| | 08/11/2011 11:09 | 11.4 | 216.4 | 10 | salt dilution |
| | 05/01/2012 14:22 | -8.7 | 4.7 | 7 | salt dilution |
| | 05/01/2012 14:26 | -8.7 | 4.7 | 7 | salt dilution |
| | 17/02/2012 13:43 | -10.2 | 2.7 | 7 | salt dilution |
| | 17/02/2012 13:46 | -10.2 | 2.7 | 7 | salt dilution |
| | 01/03/2012 14:23 | -10.6 | 2.0 | 7 | salt dilution |
| | 01/03/2012 14:27 | -10.6 | 2.1 | 7 | salt dilution |
| | 08/03/2012 16:28 | -10.4 | 1.8 | 7 | salt dilution |
| | 08/03/2012 16:33 | -10.4 | 1.8 | 7 | salt dilution |
| | 27/11/2012 06:50 | 12.5 | 210.0 | 7 | salt dilution |
| | 13/03/2013 14:55 | 0.5 | 41.9 | 7 | salt dilution |
| | 13/03/2013 14:55 | 0.5 | 40.9 | 7 | salt dilution |
| | 17/04/2013 14:40 | -3.5 | 15.1 | 10 | salt dilution |
| | 17/04/2013 14:40 | -3.5 | 15.4 | 10 | salt dilution |
| | 23/10/2013 11:00 | 31 | 1026 | 5 | salt dilution |
| | 23/10/2013 13:35 | 74.5 | 8000 | 25 | SVR |
| | 18/02/2014 15:00 | 4.5 | 105.0 | 20 | salt dilution |
| | 13/10/2014 08:20 | 7 | 169.0 | 5 | salt dilution |
| Claduègne | 01/03/2012 12:45 | 22.9 | 71.3 | 10 | salt dilution |
| | 01/03/2012 12:50 | 22.9 | 71.6 | 10 | salt dilution |
| | 08/03/2012 12:40 | 22.5 | 72.2 | 10 | salt dilution |
| | 08/03/2012 12:45 | 22.5 | 69.6 | 10 | salt dilution |
| | 06/09/2012 10:00 | 19.35 | 42.6 | 10 | current meter |
| | 10/11/2012 07:53 | 107 | 17580 | 20 | SVR |
| | 27/11/2012 08:40 | 78.5 | 7670 | 15 | SVR |
| | 17/04/2013 13:30 | 35 | 229.3 | 10 | salt dilution |
| | 17/04/2013 13:30 | 35 | 245.6 | 10 | salt dilution |
| | 13/05/2013 16:25 | 41 | 438.4 | 10 | salt dilution |
| | 13/05/2013 16:30 | 41 | 435.5 | 10 | salt dilution |
| | 31/07/2013 14:00 | 25.5 | 172.0 | 10 | current meter |
| | 23/10/2013 14:40 | 180 | 50750 | 20 | SVR |
| | 13/11/2013 14:10 | 30.5 | 298.0 | 12 | salt dilution |
| | 20/09/2014 06:48 | 73 | 5970 | 15 | SVR |
| | 10/10/2014 10:00 | 67 | 3000 | 30 | ADCP |
| | 11/10/2014 06:20 | 99 | 11890 | 20 | SVR |
| | 13/10/2014 10:50 | 60 | 1953 | 15 | current meter |
| | 13/10/2014 14:10 | 137.5 | 27800 | 15 | SVR |
| | 13/10/2014 15:20 | 121.5 | 20710 | 15 | SVR |
| | 04/11/2014 07:30 | 242.5 | 95190 | 20 | SVR |
| | 04/11/2014 09:40 | 159 | 36650 | 20 | SVR |
| | 04/11/2014 14:10 | 340 | 204000 | 25 | SVR |
| Auzon | 22/05/2013 14:15 | 21 | 1520 | 15 | ADCP |
| | 23/10/2013 11:39 | 125 | 27900 | 15 | SVR |
| | 23/10/2013 12:21 | 125 | 34700 | 15 | SVR |
| | 23/10/2013 12:25 | 130 | 55300 | 15 | SVR |
| | 04/01/2014 10:46 | 153 | 68480 | 15 | LSPIV |
| | 19/01/2014 00:00 | 170 | 280000 | 15 | Manning Strickler (IPEC) |
| | 19/01/2014 15:26 | 121 | 48480 | 15 | LSPIV |
| | 05/02/2014 11:16 | 85 | 29120 | 15 | LSPIV |
| | 06/02/2014 13:30 | 43 | 8030 | 5 | ADV |
| | 20/09/2014 07:01 | 54 | 14915 | 15 | LSPIV |
| | 11/10/2014 07:55 | 75 | 18400 | 15 | SVR |
| | 13/10/2014 13:35 | 30 | 2660 | 10 | ADV |
| | 20/10/2014 05:37 | 55 | 12200 | 15 | SVR |
| | 04/11/2014 06:21 | 177 | 86080 | 15 | LSPIV |
| | 04/11/2014 07:07 | 255 | 162000 | 15 | SVR |
| | 04/11/2014 08:31 | 282 | 169148 | 15 | LSPIV |
| | 04/11/2014 08:50 | 245 | 125000 | 15 | SVR |
| | 04/11/2014 09:01 | 261 | 146269 | 15 | LSPIV |
| | 04/11/2014 09:31 | 210 | 113968 | 15 | LSPIV |
| | 04/11/2014 09:55 | 180 | 84600 | 15 | SVR |
| | 04/11/2014 14:31 | 343 | 228860 | 15 | LSPIV |
| | 04/11/2014 14:35 | 300 | 247000 | 15 | SVR |
| | 04/11/2014 14:36 | 350 | 226000 | 15 | SVR |
| | 04/11/2014 15:05 | 365 | 257000 | 15 | SVR |
| | 04/11/2014 15:21 | 381 | 296570 | 15 | LSPIV |
| | 04/11/2014 16:21 | 317 | 206310 | 15 | LSPIV |
| | 14/11/2014 07:25 | 90 | 29200 | 15 | SVR |
| | 15/11/2014 06:31 | 103 | 38491 | 15 | LSPIV |



**Table 5: Overview of the url links and DOI that allow to access the datasets presented in this study. The datasets are organized by instrument.**

| Compartment | Instrumental device / Instrument | Op / Res | Dataset name | Mistral data access | Doi | Status within HyMeX Data and Publication Policy | Present in zip file |
|---|---|---|---|---|---|---|---|
| Rainfall | S-BAND Doppler and Polarimetric radar | Op | Operational Weather Radar ARAMIS, BOLLENE, 5min reflectivity and radial wind speed | http://mistrals.sedoo.fr/?editDatsId=705 | no | Associated scientists | no |
| | | | Operational Weather Radar ARAMIS, BOLLENE, 5min cumulative rainfall in mm | http://mistrals.sedoo.fr/?editDatsId=695 | no | Associated scientists | no |
| | | | Operational Weather Radar ARAMIS, NIMES, 5min reflectivity and radial wind speed | http://mistrals.sedoo.fr/?editDatsId=708 | no | Associated scientists | no |
| | | | Operational Weather Radar ARAMIS, NIMES, 5min cumulative rainfall in mm | http://mistrals.sedoo.fr/?editDatsId=699 | no | Associated scientists | no |
| | | | French Radar composite 5min cumulative rainfall in mm | http://mistrals.sedoo.fr/?editDatsId=703 | no | Associated scientists | no |
| | X-BAND Doppler and Polarimetric radar MXPol | Res | MXPol-EPFL-LTE Radar | http://mistrals.sedoo.fr/?editDatsId=721 | 10.14768/MISTRALS-HYMEX.721 | Public | no |
| | X-BAND fast-scanning radar WR-10X+ | Res | Le Chade LaMP X Band radar | http://mistrals.sedoo.fr/?editDatsId=796 | 10.14768/MISTRALS-HYMEX.796 | Public | no |
| | micro rain radar MRR-2 | Res | Micro Rain Radar CNRM Le Pradel | http://mistrals.sedoo.fr/?editDatsId=1110 | no | Associated scientists | no |
| | | | Micro Rain Radar LaMP Le Pradel | http://mistrals.sedoo.fr/?editDatsId=855 | 10.14768/MISTRALS-HYMEX.855 | Public | no |
| | | | Micro Rain Radar LaMP St-Étienne-de-Fontbellon | http://mistrals.sedoo.fr/?editDatsId=1112 | 10.14768/MISTRALS-HYMEX.1112 | Public | no |
| | | | Micro Rain Radar OSUG Saint-Etienne-de-Fontbellon | http://mistrals.sedoo.fr/?editDatsId=1158 | 10.14768/MISTRALS-HYMEX.1158 | Public | no |
| | | | Micro Rain Radar OSUG Montbrun | http://mistrals.sedoo.fr/?editDatsId=1159 | 10.14768/MISTRALS-HYMEX.1159 | Public | no |
| | Disdrometer Parsivel 1 | Res | DSD network, Pradel-Vignes | http://mistrals.sedoo.fr/?editDatsId=436 | OSUG data center - to be created | Public | zip1 |
| | | | DSD network, Mont-Redon | http://mistrals.sedoo.fr/?editDatsId=679 | OSUG data center - to be created | Public | zip1 |
| | | | DSD network, Pradel-Grainage | http://mistrals.sedoo.fr/?editDatsId=745 | 10.6096/MISTRALS-HyMeX.745 | Public | zip1 |
| | | | EPFL-LTE Disdrometers | http://mistrals.sedoo.fr/?editDatsId=899 | 10.6096/MISTRALS-HyMeX.899 | Public | zip1 |
| | Disdrometer Parsivel 2 | Res | DSD network, Saint-Etienne-de-Fontbellon | http://mistrals.sedoo.fr/?editDatsId=744 | OSUG data center - to be created | Public | zip1 |
| | | | DSD network, Villeneuve-de-Berg-1 | http://mistrals.sedoo.fr/?editDatsId=680 | OSUG data center - to be created | Public | zip1 |
| | | | DSD network, Villeneuve-de-Berg-2 | http://mistrals.sedoo.fr/?editDatsId=681 | OSUG data center - to be created | Public | zip1 |
| | | | DSD network, Villeneuve-de-Berg-3 | http://mistrals.sedoo.fr/?editDatsId=682 | OSUG data center - to be created | Public | zip1 |
| | rain gauge Météo-France | Op | Operational surface weather observation stations over France - Hourly data | http://mistrals.sedoo.fr/?editDatsId=627 | no | Associated scientists | zip2 |
| | rain gauge SPC Grand Delta | Op | Operational rain gauges managed by SPC Grand Delta (Berzème, Escrinet, Pont d'Ucel, Vogüe) | http://mistrals.sedoo.fr/?editDatsId=1444 | no | Public | zip2 |
| | rain gauge Hpiconet | Res | Hpiconet rain gauge network | http://mistrals.sedoo.fr/?editDatsId=656 | OSUG data center - to be created | Public | zip1 |
| | rainfall reanalysis | Res | Pluviometric reanalysis Cévennes-Vivarais | http://mistrals.sedoo.fr/?editDatsId=1183 | OSUG data center - to be created | Public | zip1 |
| Meteorology | Weather stations | Op | Operational surface weather observation stations over France - Hourly data | http://mistrals.sedoo.fr/?editDatsId=627 | no | Associated scientists | zip2 |
| | Baro-Diver | Res | limnimeter network, Gazel and Claduègne catchments | http://mistrals.sedoo.fr/?editDatsId=994 | OSUG data center - to be created | Public | zip1 |
| Surface water | Hydrometric stations | Res | Gazel and Claduègne hydro-sedimentary stations | http://mistrals.sedoo.fr/?editDatsId=993 | OSUG data center - to be created | Public | zip1 |
| | | | Acoustic Doppler Velocimeter IQ Plus, Claduègne | http://mistrals.sedoo.fr/?editDatsId=1349 | OSUG data center - to be created | Public | zip1 |
| | | | LSPIV gauging stations (Auzon hydrometric station) | http://mistrals.sedoo.fr/?editDatsId=996 | 10.17180/OBS.OHM-CV.ARDECHE | Public (access via BDOH) | zip1 |
| | Water erosion plots | Res | Runoff and erosion plots, Pradel | http://mistrals.sedoo.fr/?editDatsId=1347 | OSUG data center - to be created | Public | zip1 |
| | Stream sensors network | Res | limnimeter network, Gazel and Claduègne catchments | http://mistrals.sedoo.fr/?editDatsId=994 | OSUG data center - to be created | Public | zip1 |
| Soil | Theta Probe | Res | Soil Moisture Gazel | http://mistrals.sedoo.fr/?editDatsId=1179 | 10.6096/MISTRALS-HyMeX.1179 | Public | zip1 |
| | 10 HS | Res | Soil moisture sensor network, Gazel and Claduègne catchments | http://mistrals.sedoo.fr/?editDatsId=1350 | OSUG data center - to be created | Public | zip1 |
| | ThetaProbe ML2X | Op | SMOSMANIA - Soil moisture and temperature, France | http://mistrals.sedoo.fr/?editDatsId=469 | no | Associated scientists (access via International Soil Moisture Network) | zip2 |

**5  Table 6: Overview of the url links and DOI that allow to access the available GIS descriptors presented in this study.**

| GIS descriptor | Data | Dataset name | Data access | Doi | Status within HyMeX Data and Publication Policy | Present in zip file |
|---|---|---|---|---|---|---|
| Topography | 1 m bare earth DEM of Claduègne catchment | Digital Terrain Model (DTM) Lidar of Claduegne catchment | http://mistrals.sedoo.fr/?editDatsId=1178 | 10.6096/MISTRALS-HyMeX.1178 | Public | zip1 |
| | 5 m bare earth DEM of Auzon catchment | Digital Terrain Model (DTM) of the Auzon catchment region | http://mistrals.sedoo.fr/?editDatsId=1389 | 10.6096/MISTRALS-HyMeX.1389 | Public | zip1 |
| Soil properties | soil depth for each SCU* | Soil properties Auzon catchment | http://mistrals.sedoo.fr/?editDatsId=1385 | 10.6096/MISTRALS-HyMeX.1385 | Public | zip1 |
| | maximum soil water storage for each SCU* | Soil properties Auzon catchment | http://mistrals.sedoo.fr/?editDatsId=1385 | 10.6096/MISTRALS-HyMeX.1385 | Public | zip1 |
| | soil texture of superficial layer for each SCU* | Soil properties Auzon catchment | http://mistrals.sedoo.fr/?editDatsId=1385 | 10.6096/MISTRALS-HyMeX.1385 | Public | zip1 |
| | soil stone content for each SCU* | Soil properties Auzon catchment | http://mistrals.sedoo.fr/?editDatsId=1385 | 10.6096/MISTRALS-HyMeX.1385 | Public | zip1 |
| Infiltration tests | infiltration campaign Claduègne catchment | Infiltration campaign Claduègne catchment, Ardèche, France | http://mistrals.sedoo.fr/?editDatsId=1321 | 10.6096/MISTRALS-HyMeX.1321 | Public | zip1 |
| Land use | 5 m resolution images of Claduègne catchment | Landcover map Claduègne catchment | http://mistrals.sedoo.fr/?editDatsId=1381 | 10.14768/MISTRALS-HYMEX.1381 | Public | zip1 |
| | 30 m resolution images of Auzon catchment | Landcover map Ardeche, Cèze and Gardon Bassins | http://mistrals.sedoo.fr/?editDatsId=1377 | 10.14768/MISTRALS-HYMEX.1377 | Public | zip1 |
| Surface information | Catchment boundaries | Surface information Auzon catchment | http://mistrals.sedoo.fr/?editDatsId=1390 | 10.6096/MISTRALS-HyMeX.1390 | Public | zip1 |
| | Drainage network (stream) | Surface information Auzon catchment | http://mistrals.sedoo.fr/?editDatsId=1390 | 10.6096/MISTRALS-HyMeX.1390 | Public | zip1 |
| | Instruments | Surface information Auzon catchment | http://mistrals.sedoo.fr/?editDatsId=1390 | 10.6096/MISTRALS-HyMeX.1390 | Public | zip1 |

\* SCU means Soil Cartographic Unit.



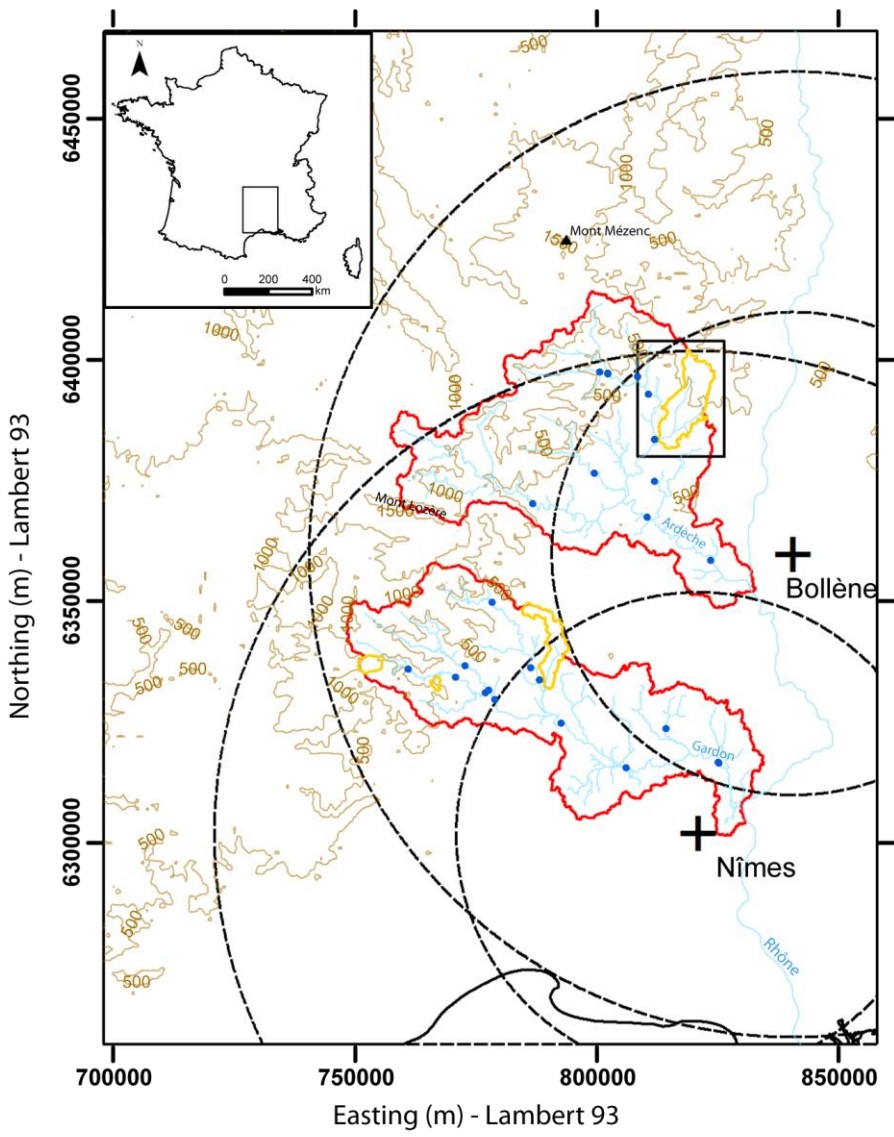

**Figure1: Location of the OHMCV pilot site. The two main catchments studied in the FloodScale project (Braud et al., 2014): Gardon (2062 km²) in the South and Ardèche (2388 km²) in the North, are outlined by the bold red line along with the main rivers and the operational hydrometric stations (blue dots). The small research catchments are shown with orange boundaries. The Auzon catchment, which is the object of this study, is framed by a black rectangle which defines the spatial extension of Figure 3. The two S-band operational radars that are of interest for this study are represented. The 500 m contour lines are displayed in the background.**



**Figure2: Typical landscapes of the Auzon catchment: (a) volcanic plateau of Coiron with a mix of grassland and open woodlands in the North part of the catchment, (b) deep valleys descending from the Coiron plateau presenting steep slopes of marls with badlands aspects and covered by deciduous forest, (c) Southern boundary of the Coiron plateau characterized by the presence of cliffs, (d) Toposequence on marly-limestone formations with regosols on steep marly slopes in the foreground followed by cultivated clayey soils with vines and ending with rocky outcrops and lithosols on limestones with garrigue, (e) Hillslopes with vineyards on clayey soils drained by a river incised in the marly-limestone bedrock and surrounded by a zone of riparian vegetation, (f) Garrigue and Mediterranean open woodland on karstified limestones in the West part of the catchment leading to the rapid drying up of the Auzon river.**



**Figure3: Location maps of the Auzon catchment and instruments for (a) rainfall, (b) meteorology, and (c) hydrology. Three different backgrounds are represented: (a) elevation (25m bare earth DEM, source: IGN), (b) land use (30 m resolution images derived from Landsat images, source: UMR Espace), (c) pedology (1:100000 soil map, source: INRA). Note that the icons used for representing the X-band radars were obtained from the IFLOODS Project Website (http://ifis.iowafloodcenter.org/ifis/more/ifloods/) presented by Demir et al. (2015).**



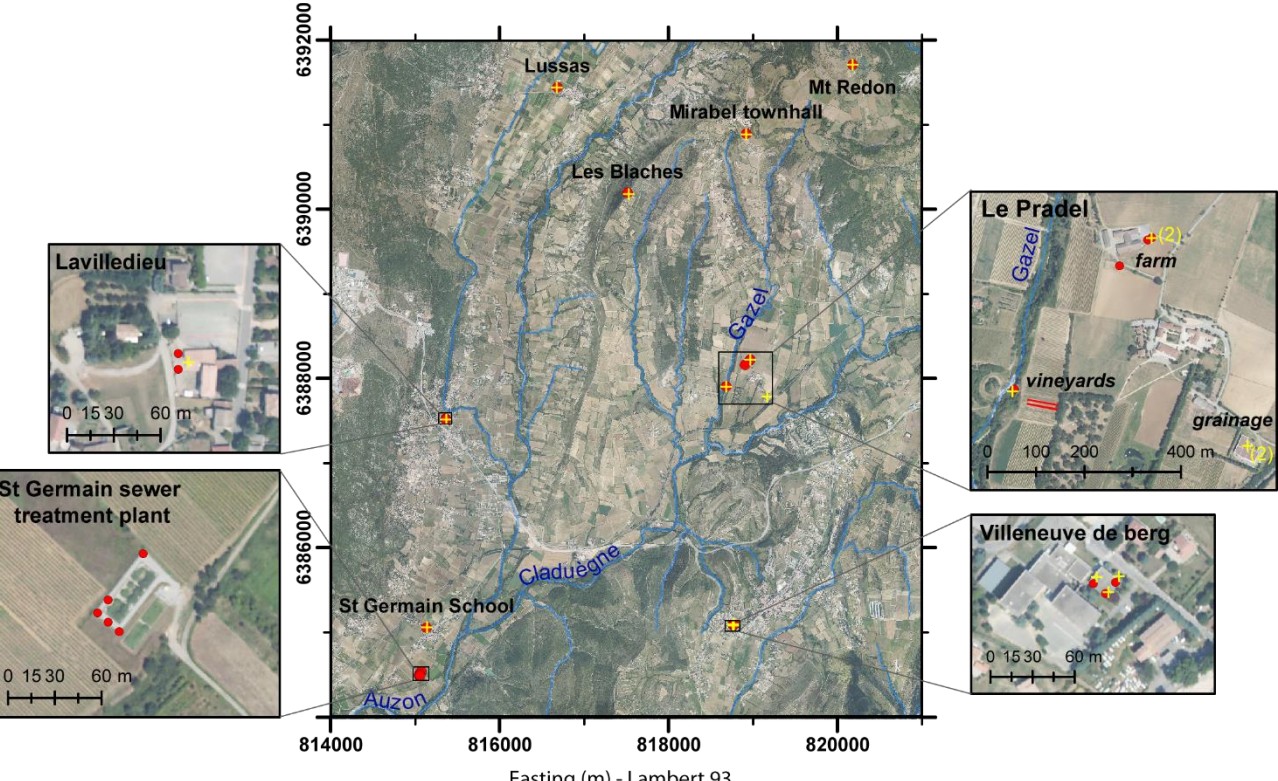

**Figure4: Location of the 19 HPicoNet rain gauges (red dots) and 14 disdrometers (yellow crosses) deployed over a 7x8 km² area. Where rain gauges are distant of less than few hundred of meters, inset maps present the configuration of the deployment at the local scale. The names of the location of the instruments are indicated in black. For "Le Pradel" site, two sub-sites are indicated and the location of the soil erosion plots is represented with red lines.**



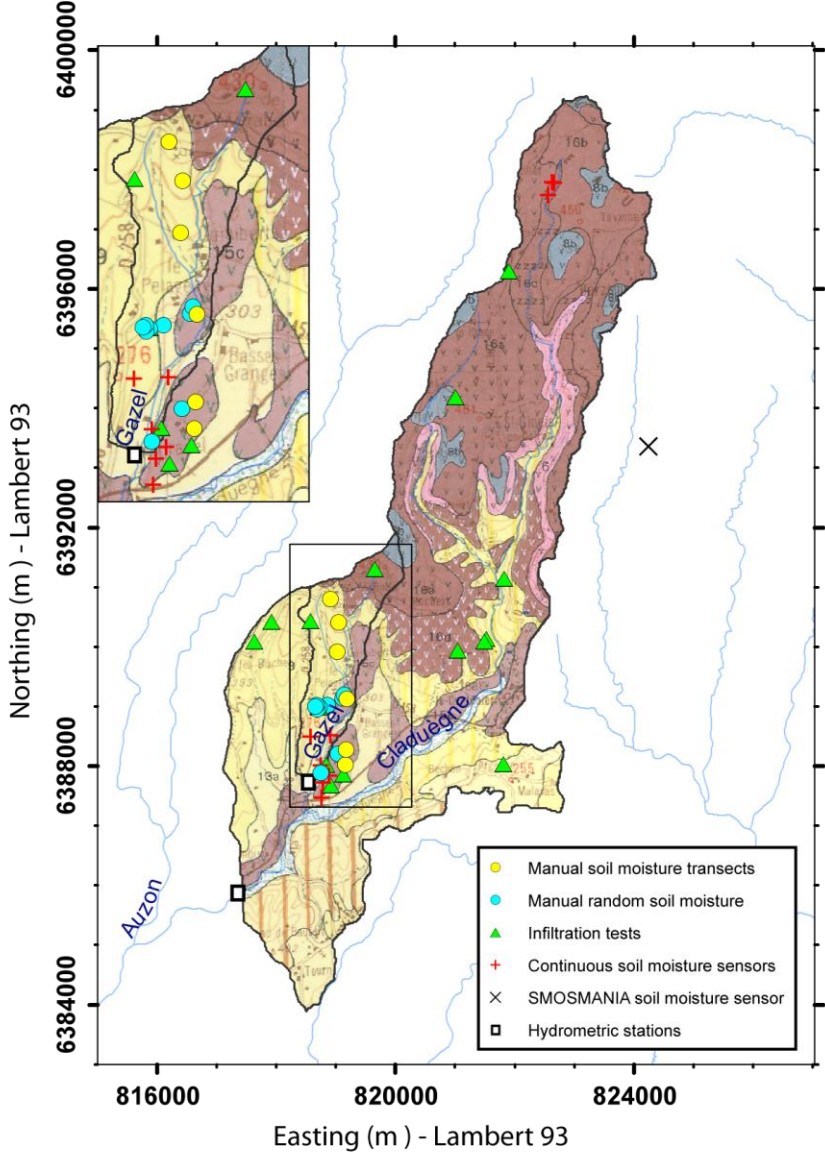

**Figure5: Location of infiltration tests and soil moisture measurements in the Gazel and Claduègne catchments. The soil moisture measurements include both manual and continuous measurements. The black rectangle shows the position of the zoom provided at the top left of the figure. The pedology (1:100000 soil map, source: INRA) is displayed in the background.**





**Figure6: Kriging with external drift estimates from radar–operational rain gauge merging (top) and ordinary kriging estimates from the operational rain gauge network (bottom) for the 04 November 2014 between 13:00 and 14:00 UTC. The graphs on the left display the hourly rain amounts (mm) and the graphs on the right display the corresponding final error standard deviations (mm).**
5    **The results are provided for a raster grid of 1 km².**



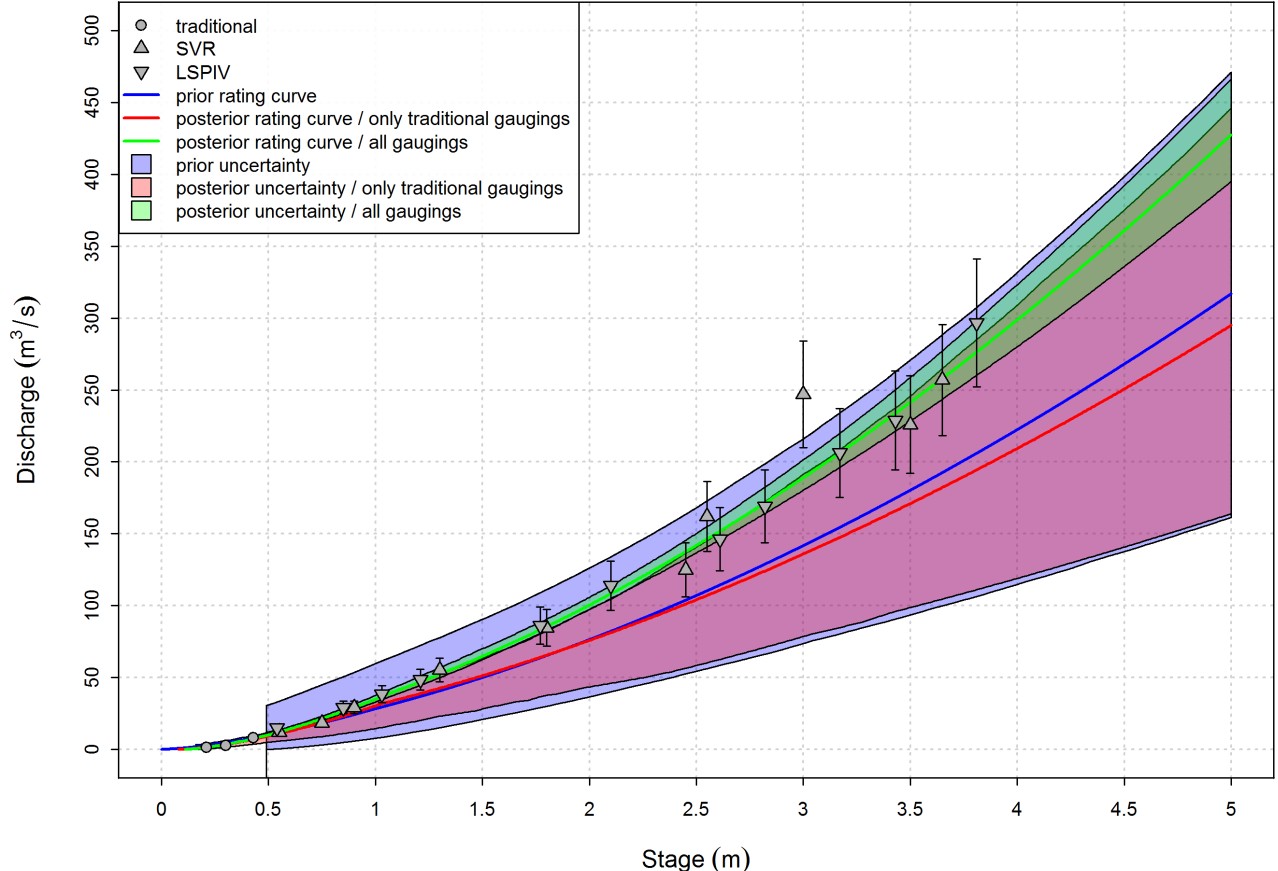

**Figure7: Stage-discharge rating curves and their uncertainty for the Auzon station: (blue) prior rating curve based on hydraulic analysis only (no gaugings); (red) rating curve established with traditional gaugings only; (green) rating curve established with all gaugings, including high flow noncontact gaugings. Solid lines represent the rating curves. Shaded areas represent the corresponding uncertainty 95% confidence intervals.**





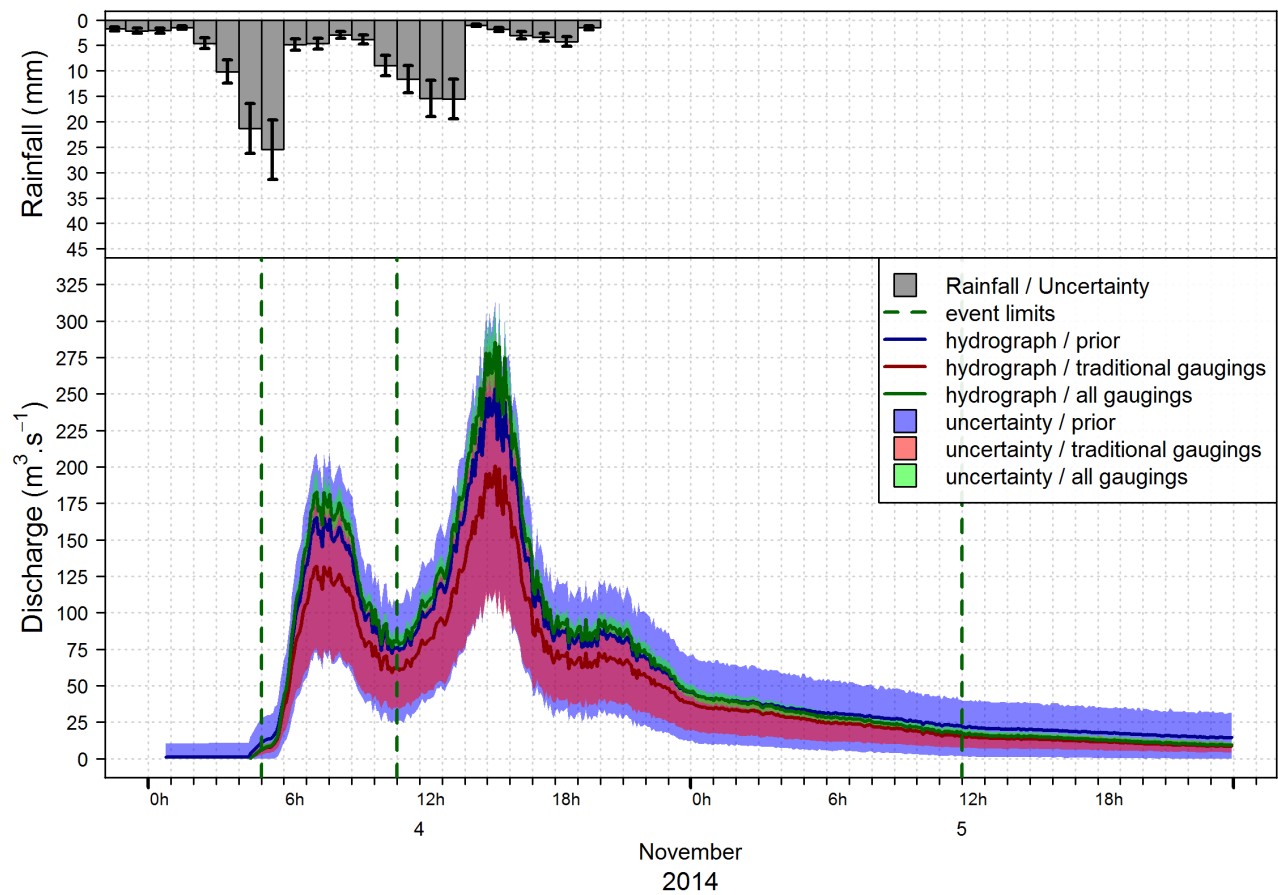

**Figure8: Event of 04 November 2014 on the Auzon catchment: hydrograph and associated uncertainty: (blue) with prior rating curve (no gaugings); (red) with rating curve established with traditional gaugings only; (green) with rating curve established with all gaugings. Solid lines represent the hydrographs. Shaded areas represent the corresponding uncertainty 95% confidence intervals.**





**Figure9: Overview of the (a) 6h accumulated rainfall, (b) Discharge (10 min time step) and (c) Turbidity (10 min time step) for the entire record (period 2011-2014) at the Claduègne hydrometric station. The rainfall data presented were taken from the operational Météo-France rain gauge "Mirabel-SA" displayed in Figure 3. Note the 4 November 2014 flood for that is a 5-10 years return period flood for the Claduègne river.**





**AppendixA: Summary of the period of operation of the instruments. The number of instruments in operation is also indicated given that many instruments belong to a network of sensors.**

| Compartment | Instrumental device | Op / Res | Instrument | 2011 | 2012 |
|---|---|---|---|---|---|
| Rainfall | | Op | S-BAND Doppler and Polarimetric radar | | |
| | | Res | X-BAND Doppler and Polarimetric radar MXPol | | |
| | | Res | X-BAND fast-scanning radar WR-10X+ | | |
| | | Res | micro rain radar MRR-2 | | |
| | | Res | Disdrometer Parsivel 1 | | |
| | | Res | Disdrometer Parsivel 2 | | |
| | | Op | rain gauge Météo-France, SPC Grand Delta | | |
| | | Res | rain gauge Hpiconet | | |
| Meteorology | | Op | Temperature probe PT100 | | |
| | | Res | Baro-Diver DI500 , Mini-Diver DI501 | | |
| | | Op | Humidity probe HMP45D | | |
| | | Op | Barometer PTB220 | | |
| | | Op | Wind sensor DEOLIA 96, Alizia 312 | | |
| | | Op | Pyranometer Kipp & Zonen | | |
| Surface water | Hydrometric stations | Res | Pressure Probe PLS | | |
| | | Res | Radar level sensor Cruzoe | | |
| | | Res | Radar level sensor RLS | | |
| | | Res | LSPIV / analogical camera VW-BP330 | | |
| | | Res | Radar surface velocity sensor RG-30 | | |
| | | Res | Acoustic Doppler Velocimeter IQ Plus | | |
| | | Res | Conductivity and Temp. Probe CS547 | | |
| | | Res | Suspended Solids probe Visolid IQ 700 | | |
| | | Res | 3700 Portable sampler | | |
| | Water erosion plots | Res | HS Flume with level sensor Thalimedes | | |
| | | Res | 3700 Portable sampler | | |
| | Limnimeter network | Res | Mini-Diver DI501 | | |
| | | Res | CTD-Diver DI271 | | |
| Soil | Soil moisture network | Res | Theta Probe | | |
| | | Res | 10 HS | | |
| | | Res | ThetaProbe ML2X | | |

| Compartment | Instrumental device | Op / Res | Instrument | 2013 | 2014 |
|---|---|---|---|---|---|
| Rainfall | | Op | S-BAND Doppler and Polarimetric radar | | |
| | | Res | X-BAND Doppler and Polarimetric radar MXPol | | |
| | | Res | X-BAND fast-scanning radar WR-10X+ | | |
| | | Res | micro rain radar MRR-2 | | |
| | | Res | Disdrometer Parsivel 1 | | |
| | | Res | Disdrometer Parsivel 2 | | |
| | | Op | rain gauge Météo-France, SPC Grand Delta | | |
| | | Res | rain gauge Hpiconet | | |
| Meteorology | | Op | Temperature probe PT100 | | |
| | | Res | Baro-Diver DI500 , Mini-Diver DI501 | | |
| | | Op | Humidity probe HMP45D | | |
| | | Op | Barometer PTB220 | | |
| | | Op | Wind sensor DEOLIA 96, Alizia 312 | | |
| | | Op | Pyranometer Kipp & Zonen | | |
| Surface water | Hydrometric stations | Res | Pressure Probe PLS | | |
| | | Res | Radar level sensor Cruzoe | | |
| | | Res | Radar level sensor RLS | | |
| | | Res | LSPIV / analogical camera VW-BP330 | | |
| | | Res | Radar surface velocity sensor RG-30 | | |
| | | Res | Acoustic Doppler Velocimeter IQ Plus | | |
| | | Res | Conductivity and Temp. Probe CS547 | | |
| | | Res | Suspended Solids probe Visolid IQ 700 | | |
| | | Res | 3700 Portable sampler | | |
| | Water erosion plots | Res | HS Flume with level sensor Thalimedes | | |
| | | Res | 3700 Portable sampler | | |
| | Limnimeter network | Res | Mini-Diver DI501 | | |
| | | Res | CTD-Diver DI271 | | |
| Soil | Soil moisture network | Res | Theta Probe | | |
| | | Res | 10 HS | | |
| | | Res | ThetaProbe ML2X | | |

Op/Res column indicates whether the instruments belong to an operational observation network ("Op") or a research observation network ("Res").

For each year from 2011 to 2014, the number of operating instruments is indicated by fortnight period (the months are numbered from 1 to 12).

10 Green colour underlines the periods when instruments operate contrary to red colour that indicates period without any measurements.