# Peer review of "A high space-time resolution dataset linking meteorological forcing and hydro-sedimentary response in a mesoscale Mediterranean catchment (Auzon) of the Ardèche region, France"

_Earth System Science Data, 2016_

## Referee Comment (RC1) · Anonymous Referee #1 · 21 Oct 2016

The manuscript submitted by Nord et al. describes a dataset for the Mesoscale Auzon catchment. The dataset contains numerous time series (starting 2011) and geodata related to meteorology, hydrology and geomorphology/soil sciences. Most data was acquired within the HYMEX project and have already enabled numerous scientific publications.

general comments ————— The submitted manuscripts describes a dataset of impressive extent: To my knowledge, the associated collection is unique in sheer size, detailedness and aspects covered for a catchment at this scale. Numerous studies

have already built upon this data; opening it to the public will doubtlessly maximize the scientific payoff of the measurement efforts. While some parts of the data may have been available before (e.g. via MISTRAL database and related DOIs, Llasat et al. 2013a,b), the well-written manuscript creates a thorough overview of the available information. The data is mainly stored within the well-designed MISTRAL database in well-documented and common formats. The suggested* "bundled data" further facilitates the already quite convenient access for the reader. I am convinced that the scientific community will gratefully accept this valuable contribution. While I suggest (mainly cosmetic changes) to the manuscript itself, I ran into several issues concerning the accessibility, completeness and consistency of the data. None of these do severely impair the scientific value, but will still require amendments. Since I could not access many referenced files, subsequent revisions may reveal further issues in these. I recommend moderate revisions.

specific comments —————— (The annotated PDF contains numerous, mostly minor comments and details for the points raised here.) 1. The number of authors seems unduly high. However, considering the tremendous measurement effort involved it MIGHT still be warranted. The editor should make a decision on this. 2. The manuscript is relatively long (39 pages). ESSD states that "Long articles are not expected." However, apart from the introduction (see below), the examples (see below) and some figures I could not identify any obsolete content. 3. intro could be shortened and better structured. Suggestion: identify three main potential usage groups for the dataset (i.e. process understanding, model development/testing, management support). Then briefly outline concrete scientific questions/studies for these categories. The current line of thought is somewhat erratic. 4. The manuscript claims "All the data presented here have undergone careful (mostly manual) quality assurance." It would be helpful to elaborate, wherever possible, what exactly this means, ie what aspects/error sources were screened. Since I ran into several flaws by random checks, this needs to be clarified. 5. Section 4 gives examples of use of the data. While these examples are certainly interesting, it seems there are also numerous other studies
based on the dataset. Thus, instead of picking examples, an overview table listing the key questions addressed using the data and the corresponding publications would be more useful and streamline the paper. If, however, this section is to be remain, some details of the examples need better explanations. 6. minor issues about soil moisture measurement description 7. The stream gauges feature multiple redundant systems. It is not fully clear, which of the reported devices were used in the end. 8. I could not find any explicit statement on this, but it appears that at least some of the instruments are still running. This should be stated, wherever possible. In that context, the MISTRAL website suggests that the respective data is still being supplemented. On the one hand, further extending the data is certainly useful. On the other hand, in the context of reproducibility and benchmarking, it needs to be stated if and how "old" versions of the data are preserved (example: Scientist XY uses discharge data for a study, HYMEX team later performs more Q-mesurements and updates hQ-relationship and Q-series.) 9. Exhaustively scrutinizing the provided dataset is beyond my time and knowledge limits. Instead, I have sampled 21 features listed in Tables 5 and 6. Of these, I could not find any flaws for 2, but ran into multiple access and data consistency issues for the others (for details see test_access.xlsx). Similar (or other) issues may also affect the dataset I did not check. General issues: a) information on timezones (UTC, CET, CEST?) should be provided wherever possible *b) Providing "bundled data" seems to be a very useful idea. I could not abstract how to obtain these bundled data in the respective zip files, though. c) turbidity data is more useful when SSC-samples are provided d) Some data mentioned in the manuscript are apparently not part of the publication (e.g. geology map). It should be made clear wherever this is the case. e) The provided information on publication policy is misleading. From the entries I tested, only one required no registration and was truly public. Conversely, many datasets labeled as "Associated scientist" were completely inaccessible for me.

Please also note the supplement to this comment:
http://www.earth-syst-sci-data-discuss.net/essd-2016-32/essd-2016-32-RC1-

supplement.zip
* * *

---

## Referee Comment (RC2) · Anonymous Referee #1 · 24 Oct 2016

Another thing that needs to be discussed is the policy for data access and data re-use. Concerning access, I already expressed some concerns. More specifically, it needs to be made clear how datasets are *accessible* and under what conditions the data can be *reused*:

access: * truly public (ie. directly accessible) * after registration * with additional costs (i.e. cannot be part of this publication)

reuse: * free (i.e. Public Domain, Creative Commons or similar) * additional restrictions

Specifically, I would like to discuss the implications of the HYMEX data policy

(http://mistrals.sedoo.fr/HyMeX/Data-Policy/HyMeX_DataPolicy.pdf), which seems to govern the largest part of the data.

a) Section 3 §4 states that data can be made public domain. However, imposing additional obligations (section 4) contradicts to the common notion of PD as being completely free. b) Section 4 §2 restricts the use of the data to objectives of the HYMEX programme. c) While the correct attribution of the data being used of course is scientific best practice (section 4, §5), the license also requires to offer collaboration to the PI (§6) and co-authorship (§7). d) Derived data and publications need to be provided to HYMEX (§11-13).

In my opinion, a) needs clarification. b) likely restricts the use for some scientific research questions and should be removed. c) with §6 and §7 is ethically undue. The authors of the data have already claimed their academic rewards by (then) having published the ESSD-paper and it being cited in case of re-use. While any researcher may be happy to contact the PI and invite him to collaborate, it cannot be mandatory. d) loosely corresponds to a Creative Commons Attribution-NonCommercial 4.0 International (CC BY-NC 4.0), which is acceptable.

These are, of course, ethical decisions and thus open to debate. The requirements of ESSD are quite diffuse about that (data needs to be "openly accessible (cost-free for readers)"). I specifically ask the other reviewers and the editor to express their opinion on these points.
* * *

---

## Referee Comment (RC3) · Anonymous Referee #2 · 8 Nov 2016

This work describes a comprehensive dataset that includes hydrometeorological, and hydrological data collected from a number of sensors (in situ and remote) over a mesoscale catchment in Mediterranean. The number of observed variables, the various sensors involved and the space/time resolution of this comprehensive dataset makes it a unique contribution to research community. I agree with the authors that such a dataset can serve as an excellent benchmark for evaluating and improving process based models used for understanding triggering rainfall properties, runoff generation mechanisms and erosion processes during Mediterranean floods.

[Figure]

My recommendation is to accept the paper for publication in ESSD journal. Below I list only some minor points/corrections that may help to improve information provided in some specific parts of the text. However, I want to share also the concerns raised by reviewer#1 regarding the conditions that may potentially apply on the use of this dataset. My understanding from reading the manuscript is that the "non public" data require simply a registration to HyMeX before they can be retrieved and used. If there is more into this (see points/concerns of reviewer#1) it needs to be clarified and dataset needs to be subsequently adjusted to include only data that are unconditionally available.

Minor points

1. P1L29: "...quantity and type of sensors", consider improve wording 2. P2L4: "...measure water discharge...", do you mean water stage or "estimate water discharge"? 3. P2L26: "Indeed, the water and sediment discharges simulated by distributed models...are generally poorer...". It is stated as a universal truth. I suggest to revise statement to mention that "many studies have shown...", for example. 4. P2L28: "this raises the question of the improvement...." Improve wording/syntax. 5. P3L6: "...to exceed the current limitation..." consider changing this to "to improve distributed models" or "to address current limitations" 6. P3L16: I am not sure about the "leaser extent on-board satellites...". What is the basis for stating that? Especially since you are referring to rainfall all around the world where in many parts satellites are the only source of information. 7. P3L21: "rainfall is not steady..." what do you mean by stead rainfall? 8. P3L22: "..the scales of operational rain gauge...". I am not sure what is your reference here but at a global level the typical temporal scale of rain gauge observations is daily. 9. P3L24: "operational radars is essential", for what? The sentence ends rather abruptly. 10. P4L25: Have you previously define what SOP and EOP stands for? 11. P7L19:which one is the MXPol? Is it the EPFL-LTE? It is a bit confusing. 12. P8L12:"..is made thanks to a Hobo..." consider revising and avoiding "thanks to" 13. P14L27: Not clear what you mean by "radar-rain gauge" here.
Do you refer to a merging technique? OR just a geostatistical-based interpolation to obtain "radar-like" QPE from gauges. Please clarify. 14. P15L31: "...we saw during the events of the 2014", how did you see that? Please explain. 15. P16L12-13: NSE and PBIAS are not defined previously. 16. P16L22: "5-10 yr return period". This is interesting and I am wondering what is the basis for this estimate? Are there any flood frequency curves available for the area? It would be interesting to have this infor for other flood events in the record. 17. P17L28: "...quality (size, fall velocity...", please revise because size and velocity of hydrometeors is not a qualitative measure.

---

## Short Comment (SC1) · 17 Nov 2016

Dear reviewer, We thank you for your careful reading of the paper. Before replying point by point to your comments, I realize that a lot of your comments are related to data accessibility. So, I have a general question: were you able to download the bundled data? I mean the files "zip1_auzon.zip" and "zip2_auzon.zip" available from the link to the data repository included in the abstract of the paper (http://mistrals.sedoo.fr/MISTRALS/?editDatsId=1438). Access to the file "zip1_auzon.zip" is totally public (see the flag "Direct access to public data") and does

not require any identification. Access to the file "zip2_auzon.zip" is subject to the HyMeX data policy. We created a special account to allow publishers and reviewers to download this file "zip2_auzon.zip". We had sent a "cover letter" to the publisher at the time of submission of the paper to explain the procedure to download all the data (in particular the username and password of the account that enable to download the file "zip2_auzon. Zip "). Have you received this explanatory note? This bundling service represented an important task. It was performed to ease the download tasks for the interested users (as described in Section: 5 Data availability) as it enables to avoid the download of each individual datasets (41 in total). Best regards.

―――――――――――――――――――

Fig. 1.

---

## Author Comment (AC1) · 16 Dec 2016

**Final author comments to referee 1 comments on "A high space-time resolution dataset linking meteorological forcing and hydro-sedimentary response in a mesoscale Mediterranean catchment (Auzon) of the Ardèche region, France" by Nord et al.**

In the following, the reviewer comments appear in black italic and our answers are provided in blue. When there are quotations from the text of the article, they appear in quotation marks and the new or corrected parts are highlighted in yellow.

essd-2016-32-RC1.pdf

*The manuscript submitted by Nord et al. describes a dataset for the Mesoscale Auzon catchment. The dataset contains numerous time series (starting 2011) and geodata related to meteorology, hydrology and geomorphology/soil sciences. Most data was acquired within the HYMEX project and have already enabled numerous scientific publications.*

*general comments —————- The submitted manuscripts describes a dataset of impressive extent: To my knowledge, the associated collection is unique in sheer size, detailedness and aspects covered for a catchment at this scale. Numerous studies have already built upon this data; opening it to the public will doubtlessly maximize the scientific payoff of the measurement efforts. While some parts of the data may have been available before (e.g. via MISTRAL database and related DOIs, Llasat et al. 2013a,b), the well-written manuscript creates a thorough overview of the available information. The data is mainly stored within the well-designed MISTRAL database in well-documented and common formats. The suggested\* "bundled data" further facilitates the already quite convenient access for the reader. I am convinced that the scientific community will gratefully accept this valuable contribution.*

Answer: We thank referee 1 for this positive appraisal of the dataset and the paper content. We would like to mention that we have embarked on this project of publication in order to present the whole data set, make it more accessible and increase its potential for use and encourage new studies. We consider that current studies based on the dataset are not so numerous to date but we hope that this will lead to more publications in the future and by new teams.

*While I suggest (mainly cosmetic changes) to the manuscript itself, I ran into several issues concerning the accessibility, completeness and consistency of the data. None of these do severely impair the scientific value, but will still require amendments. Since I could not access many referenced files, subsequent revisions may reveal further issues in these. I recommend moderate revisions.*

Answer: We thank sincerely referee 1 for the careful reading of the paper and the thorough inspection of the dataset which represent a very valuable work. We have taken into account the issues concerning the accessibility, completeness and consistency of the data highlighted by referee 1. We have performed a complete inspection about the accessibility of the data (see comment. 9). Concerning the completeness and the consistency of the data, the corrections are detailed later on.

*specific comments  —————  (The annotated PDF contains numerous, mostly minor comments and details for the points raised here.)  1.  The number of authors seems unduly high. However, considering the tremendous measurement effort involved it MIGHT still be warranted. The editor should make a decision on this.*

Answer: The dataset presented in this paper is very large which may explain the long list of authors. It appeared to us that it made more sense and the scope was greater if the dataset was published as a whole instead of splitting into several publications. In return, the consequence is that the article has many authors. The contribution of the different authors was explained in the section "Author contribution". We would like to know the editors' position on this issue.

*2. The manuscript is relatively long (39 pages). ESSD states that "Long articles are not expected." However, apart from the introduction (see below), the examples (see below) and some figures I could not identify any obsolete content.*

Answer: Once again, the paper describes a very extensive and therefore uncommon dataset. It appeared to us that it made more sense and the scope was greater if the dataset was published as a whole instead of splitting into several publications. In return, the consequence is that the article is perhaps longer than the ordinary. As referee 1 does not identify any obsolete content, we are not in favor of a significant reduction in paper length. Nevertheless, we agree with a reduction and re-organization of the introduction (see below) and we propose to move Figure 9 to Appendix B.

*3. intro could be shortened and better structured.  Suggestion:  identify three main potential usage groups for the dataset (i.e.  process understanding,  model development/testing,  management support). Then briefly outline concrete scientific questions/studies for these categories. The current line of thought is somewhat erratic.*

Answer: We are aware of the improvements that can be made to the introduction. We thank referee 1 for the propositions. We have completely re-written and shortened the introduction. It now reads as follows:

"The Mediterranean area is prone to intense rainfall events, sometimes triggering flash floods that may have dramatic consequences (Ruin et al., 2008). Flash floods are the consequence of short, high-intensity rainfalls mainly of spatially confined convective origin and often enhanced by orography (Borga et al., 2014). As such, flash floods usually impact basins less than 1000 km² (Marchi et al., 2010). In medium-scale Mediterranean catchments, the control exerted by the amount of rainfall and its intensity and variability on the generation of runoff and the erosional processes operating at different scales is of major importance (Navratil et al., 2012; Marra et al., 2014; Tuset et al., 2015). Assisting stakeholders in implementing efficient soil conservation and river management measures implies understanding the processes and the factors that control surface runoff, develop modelling approaches able to provide reliable flow separations, localize sediment sources and sinks, and predict the space-time dynamics of sediment and associated contaminant within the catchment. This requires taking into account the space-time variability of rainfall events, using spatially distributed models coupling hydrology and mass transfers.

Although the interest of distributed models is recognized for understanding the inner behaviour of the catchment (i.e. pathways and transit times), many studies have shown that their reliability do not meet the expectations. Indeed, the water and sediment discharges simulated by distributed models

at the outlet of the catchment are generally poorer than the results simulated by lumped models (Jetten et al., 2003; Reed et al., 2004; de Vente el al., 2013). To date there are various difficulties that hinder the potential of distributed models (e.g. Cea et al., 2016) such as the large number of parameters, the definition of some parameters which are difficult to measure, the high non-linearity of the equations, the interaction between input parameters, the uncertainty in the experimental measurements and input data, the space-time variability of the physical processes, and the lack of comprehensive field data available for initialization and calibration. Thus the deployment of multi-scale observation systems over a period of several years in medium catchments and the release of the collected datasets as open data with metadata on how the data have been collected, quality assured, and their associated uncertainties (Weiler and Beven, 2015) is of crucial importance to address the current limitations of distributed models.

High space-time resolution (~ km², ~ min) datasets linking meteorological forcing and hydro-sedimentary response are rare in scientific literature because of the high number and diversity of types of sensors required for measuring rainfall and surface hydrology. The already published datasets consist of first order catchments ("Tarrawarra data set" (southeastern Australia): Western and Grayson, 1998), catchments where the observation period is exceptionally long ("Reynolds Creek Experimental Watershed" (northwestern USA): Slaughter et al., 2001; "Walnut Gulch Experimental Watershed" (southwestern USA): Renard et al., 2008; Stone et al., 2008; "Goodwater Creek Experimental Watershed and Salt River Basin" (midwestern USA): Baffaut et al., 2013), or catchments located in snow-dominated mountain ("Reynolds Creek Experimental Watershed" (northwestern USA):  Reba et al., 2011; "Dry Creek Experimental Watershed" (northwestern USA): Kormos et al., 2014). In mesoscale catchments, such datasets are scarce ("Walnut Gulch Experimental Watershed" (southwestern USA): Goodrich et al., 1997; "Iowa River Basin" (north central USA): Gupta et al., 2010), especially in the Mediterranean region.

This study is part of the FloodScale project (Braud et al., 2014), which is a contribution to the HyMeX program (Hydrological Cycle in the Mediterranean Experiment, Drobinski et al., 2014), a 10-year multidisciplinary program on the Mediterranean water cycle. A three-level nested experimental strategy was planned for the HyMeX program:

- A long-term observation period (LOP) lasting about 10 years (2010-2020) to gather and provide observations of the whole coupled system that support analysis of the seasonal-to-interannual variability of the water cycle through budget analyses.

- An enhanced observation period (EOP) lasting about 5 years (2011-2015), for both budget and process studies.

- Special observation periods (SOP) of several months, which aimed at providing detailed and specific observations to study key processes of the water cycle in specific Mediterranean regions, with emphases put on heavy precipitation systems and intense air-sea fluxes and dense water formation.

The FloodScale project (2012-2015) fits into the EOP and encompasses the SOP1 (Ducrocq et al., 2014) which took place from 5 September to 6 November 2012 and was dedicated to heavy precipitation and flash-floods. This study focuses on nested scales that range from the hillslope to the medium catchment scale all belonging to the Cévennes - Vivarais Mediterranean Hydrometeorological Observatory (OHMCV) (Boudevillain et al., 2011). It is located in Ardèche, in a region with a high gradient in annual rainfall (e.g. Molinié et al., 2012). The observation system has been operated by different teams from various countries during the SOP1 and EOP: LTHE, IRSTEA Lyon, EPFL, Wageningen University, LAMP and Météo-France. The dataset includes precipitation and

weather data, soil moisture data, runoff and soil erosion data, hydrologic and suspended sediment response data, surface water quality data, and GIS data.

The duration of the observations presented here (4 years, from 1 January 2011 to 31 December 2014) allows the characterization of the standard catchment behaviour and provides the opportunity to observe less ordinary events with processes that are specific to flash floods and to characterize possible threshold effects that are not observed in small to moderate events. The observation strategy is reinforced by the deployment of conventional and polarimetric radars that provide precipitation measurements at spatial scales not properly resolved by rain gauges networks (Berne and Krajewski, 2013). A special effort was dedicated to soil moisture measurements and stream gauging during floods. These opportunistic observations made possible by a real-time warning system enable to watch transient processes like runoff, to monitor the increase of water content in soil and to gauge high discharges in small to medium catchments, which is challenging due to the very short response times of such systems. This allows documenting the upper ends of stage-discharge rating curves that are generally extrapolated at high values.

The paper presents the acquired datasets to make them accessible to the scientific community and make their use easier and wider. The authors are convinced that the published datasets can serve as a benchmark for hydrological distributed modelling applied to the Mediterranean area. The paper is organized as follows: Section 2 presents the location of the studied catchment and its context (geology, climatology, land use, pedology). Section 3 describes the observation system (instruments and measured variables) and is organized in three subsections: i) hydrometeorological data, ii) spatial characterization data, iii) hydrological and sediment data. Finally, in Section 4, the first studies that provide preliminary answers to the scientific questions selected in the introduction are highlighted."

*4. The manuscript claims "All the data presented here have undergone careful (mostly manual) quality assurance." It would be helpful to elaborate, wherever possible, what exactly this means, ie what aspects/error sources were screened. Since I ran into several flaws by random checks, this needs to be clarified.*

Answer: Indeed, in the introduction part of section 3 (p.6 l.26-27), we wrote "All the data presented here have undergone careful (mostly manual) quality assurance." Then the procedure of data quality control is described in each sub-section. The remark made here is not very precise. We have corrected or completed the flaws when they were identified by referee 1 as specified below.

*5. Section 4 gives examples of use of the data. While these examples are certainly interesting, it seems there are also numerous other studies based on the dataset. Thus, instead of picking examples, an overview table listing the key questions addressed using the data and the corresponding publications would be more useful and streamline the paper. If, however, this section is to be remain, some details of the examples need better explanations.*

Answer: We have embarked on this project of publication in order to present the whole data set, make it more accessible and increase its potential for use and encourage new studies. Current studies based on the dataset are not yet very numerous to date. In section 4, we decided to present the first studies that provide preliminary answers to the scientific questions selected in the introduction:

1) The control exerted by the amount of rainfall and its intensity and variability on the generation of runoff and the erosional processes operating at different scales

2) The uncertainty in the experimental measurements with a specific focus on documenting the upper ends of stage-discharge rating curves that are generally extrapolated at high values

3) The use of spatially distributed models coupling hydrology and mass transfers with comprehensive field datasets to address their current limitations.

Therefore the title of section 4 was modified: "Example of data use" was replaced by "First applications using the dataset". Furthermore, a few sentences were added at the end of the introduction to explain the organization of the paper and to announce in particular the role of section 4. It now reads as follows: "The paper is organized as follows: section 2 presents the location of the studied catchment and its context (geology, climatology, land use, pedology). Section 3 describes the observation system (instruments and measured variables) and is organized in three subsections: i) hydrometeorological data, ii) spatial characterization data, iii) hydrological and sedimenta data. Finally, in section 4, the first studies that provide preliminary answers to the scientific questions selected in the introduction are highlighted."

*6. minor issues about soil moisture measurement description*

Answer: We believe that referee 1 refers to the comments that were made in the annotated version of the paper for section 3.3.2. The corresponding changes are described later on.

*7. The stream gauges feature multiple redundant systems. It is not fully clear, which of the reported devices were used in the end.*

Answer: There are many instruments used in this study. We tried to be as clear as possible when we presented the observation system. Nevertheless, we understand the difficulties of clarity which may have remained in the first version of the document. In Table 2, the choice was made to classify by variable type rather than instrument location. We propose to conserve the same organization but add a column for the location of the instruments in Table 2.

*8. I could not find any explicit statement on this, but it appears that at least some of the instruments are still running. This should be stated, wherever possible. In that context, the MISTRAL website suggests that the respective data is still being supplemented. On the one hand, further extending the data is certainly useful. On the other hand, in the context of reproducibility and benchmarking, it needs to be stated if and how "old" versions of the data are preserved (example: Scientist XY uses discharge data for a study, HYMEX team later performs more Q-mesurements and updates hQ-relationship and Q-series.)*

Answer: It is true that we have focused on the period 2011-2014 in this paper whereas we have not talked about the adjacent periods. This is a flaw that we have corrected. In the introduction, we have added the definition of the different HyMeX observation period (LOP, EOP, SOP) by stating that the period 2011-2014 belongs fully to the EOP phase. It now reads as follows:

"A three-level nested experimental strategy was planned for the HyMeX program:

- A long-term observation period (LOP) lasting about 10 years (2010-2020) to gather and provide observations of the whole coupled system that support analysis of the seasonal-to-interannual variability of the water cycle through budget analyses.

- An enhanced observation period (EOP) lasting about 5 years (2011-2015), for both budget and process studies.

- Special observation periods (SOP) of several months, which aimed at providing detailed and specific observations to study key processes of the water cycle in specific Mediterranean regions, with emphases put on heavy precipitation systems and intense air-sea fluxes and dense water formation."

Some of the instruments were uninstalled at the end of 2015 while another part was left to document the LOP. This information indicating on going measurements has been added to Table 5 and Appendix A by placing an asterisk on each line when the instrument is still running.

The data for the years 2015 and 2016 are progressively uploaded to the HyMeX database once the quality control procedure is completed. This is the case, for example, for the Hpiconet rain gauge network dataset: http://mistrals.sedoo.fr/MISTRALS/?editDatsId=656.

Referee 1 also mentions the versioning of the individual datasets, which represents a quality procedure in the long term life of a dataset. This is an important recommendation when creating a doi for a dataset. The different versions are indicated in the metadata and available either directly on the specific landing page of the dataset (the landing page is necessarily associated to the doi of the dataset) or on request from managers of the HyMeX database. Furthermore, the creation of an account in the HyMeX database even for downloading "public data" makes possible to trace the users of the data and contact them in case of an important updating of the datasets.

*9. Exhaustively scrutinizing the provided dataset is beyond my time and knowledge limits. Instead, I have sampled 21 features listed in Tables 5 and 6. Of these, I could not find any flaws for 2, but ran into multiple access and data consistency issues for the others (for details see test_access.xlsx). Similar (or other) issues may also affect the dataset I did not check.*

Answer:

Multiple access issues

Once again, we thank referee 1 for the thorough inspection of many individual datasets listed in Tables 5 and 6. The comments allowed us to correct some problems of access to the individual datasets using the links presented in Tables 5 and 6. For example, the following individual datasets all presented the same problem:

http://mistrals.sedoo.fr/?editDatsId=721
http://mistrals.sedoo.fr/?editDatsId=796
http://mistrals.sedoo.fr/?editDatsId=855
http://mistrals.sedoo.fr/?editDatsId=1112
http://mistrals.sedoo.fr/?editDatsId=1158
http://mistrals.sedoo.fr/?editDatsId=1159
http://mistrals.sedoo.fr/?editDatsId=1381
http://mistrals.sedoo.fr/?editDatsId=1377

Although the status of the datasets was "public", the following error was always encountered "ERROR, The requested URL could not be retrieved". This error has been corrected. We recommend to use Mozilla Firefox if the data are downloaded from a web browser (possible problems with Google Chrome for example). Alternatively, the FTP server can be connected using a FTP client such as FileZilla.

The most frequently encountered problem was formulated as follows by referee 1: "Requires MISTRAL credentials despite being labelled as public." Indeed the HyMeX database requires creating an account even for a user who wants to download only data under public status. We recognize that this step certainly constitutes a complication for a fully open and prompt access to the data. However, it also makes possible to trace the users of the data and contact them in case of an important updating of the datasets. The main problem was that this creation of account required the user to validate the HyMeX data policy whereas the data under public status are not subject to this policy (indeed the HyMeX data policy only applies to data under "associated scientists" or "core users" status). This was an error and was not ethically founded. As a result, with the help of the managers of the HyMeX database, we have modified the step of creation of account by this way (http://mistrals.sedoo.fr/User-Account-Creation/): users wanting to download only datasets under public status no longer need to validate the HyMeX data policy. A simple registration (name, affiliation, e-mail) is necessary to create an account (an e-mail with their login and password is instantaneously sent to their e-mail address) and then access directly to public data.

[Figure]

If users want to use datasets under "associated scientists" status, two situations are possible:

1) Either they have registered previously for access to public data only. In this case, they must go to their account (http://mistrals.sedoo.fr/Your-Account/) and complete the HyMeX registration application.
2) Or they register for the 1st time. They must go to http://mistrals.sedoo.fr/User-Account-Creation/ and do not check the box corresponding to "Simple registration (name, affiliation, country) for direct access to public data only."

In both cases, they will arrive on a page where they have to describe the planned work in more than 350 characters and they have to validate the HyMeX data policy by accepting various statements. A sentence has been added at the foot of the page to explain the procedure of validation: "Registrations are handled by the MISTRALS Executive Comittee following the criteria established by the HyMeX International Scientific Steering Committee. Registration is reviewed within 7 days and is valid for 3 years." The database managers have calculated that, since the creation of the HyMeX database, the average period of validation for applications is: 4.43 days.

Most of the individual datasets (32 of 41) have "public" access. A few of them (9 of 41) are under the "Associated scientists" status of the HyMeX data policy.

Nevertheless, to facilitate the use of the data and to avoid downloading each individual datasets (41 in total), a bundling service was provided. The bundled data presents the advantage of gathering data in ASCII and cartesian format, in a single coordinate system, and in the same timezone (UTC). The bundled data were selected for the spatial and temporal windows presented in the paper since some individual datasets have different extents. It represented an important task. As explained in section 5 Data availability, the bundled data were organized in two zip files: the "zip1_auzon.zip" and "zip2_auzon.zip" files available from the link to the data repository included in the abstract of the paper (http://mistrals.sedoo.fr/MISTRALS/?editDatsId=1438).

Access to the file "zip1_auzon.zip" is totally public (see the flag "Direct access to public data" in the page http://mistrals.sedoo.fr/MISTRALS/?editDatsId=1438) and does not require any identification. The data can be downloaded directly without any step of identification. In contrast, access to the file "zip2_auzon.zip" is under "associated scientists" status and is subject to the HyMeX data policy. We created a special account to allow editors and reviewers to download this file "zip2_auzon.zip". We had sent a "cover letter" at the time of first submission of the paper to explain the procedure to download all the bundled data (in particular the username and password of the account that enable to download the file "zip2_auzon.zip ").

data consistency issues (see test_access.xlsx)

*http://mistrals.sedoo.fr/?editDatsId=436* : *Folder contains data from 5.9.-5.11.2012, contrasting data availability specified in Table A. For the tested day (5. Sep 2012), the file contained only station 33, opposed to 9 stations claimed in Table A.*

Answer: The data were updated for the whole period. Concerning the number of stations indicated in Table A, we confirm that the value is correct. We added the following information in Table 5 for EPFL-LTE Disdrometers: "at least 6 for Fall 2012 and Fall 2013". Anyway, the simplest way to access all the disdrometers data is to download the bundled data and look at the following directories:
…\zip1_auzon\zip1\rainfall\disdrometer Parsivel 1\DSD-EPFL-30SEC-CSV
…\zip1_auzon\zip1\rainfall\disdrometer Parsivel 1\DSD-LAMP-30SEC-CSV
…\zip1_auzon\zip1\rainfall\disdrometer Parsivel 1\DSD-LTHE-1MIN-CSV

*http://mistrals.sedoo.fr/?editDatsId=656* : *Opposed to Table A, the dataset does not contain Dec 2014. Also, two stations (Lussas-step, Mirabel-Pradel-Ferme-1) do not contain any valid data, reducing the number of valid time series to 20. The station Mirabel-Mont-Redon apparently did not record from 15 till 29 Sep 2014. The respective datavalues need to be masked as "nodata".*

Answer: We agree with referee 1: the dataset does not contain December 2014. The number of instruments in operation has been updated in Table Appendix A for the entire period 2011-2014. It can be seen that there are at most 20 rain gauges that operate simultaneously. Therefore, this value has also been also modified in Table 1.

*http://mistrals.sedoo.fr/?editDatsId=656* : *multiple strange time gaps (e.g. from 201101062300 to 201101090000). The file contains only 1728 of the expected 8760 hourly values for that year (< 20 %). Apparently unmasked nodata values (e.g. Saint-Germain-Ecole 20.-30.12.2010)*

Answer: The low number of records in the dataset relatively to the number of five-minute slots in the observation period is related to two factors. One is the occurrence of malfunctioning periods which is indicated in the dataset using the NaN flag. The other one is that days where none of the 21 rain gauges record rainfall, are not in the data file in order to save memory space.

*http://mistrals.sedoo.fr/?editDatsId=994* : *File contains data only for baro2, opposed to table A listing 2 time series. Also, the data does no start before 26.Nov 2012.*

Answer: the verification was carried out on 24/11/2016 both in the file "zip1_auzon.zip" and in the data available at http://mistrals.sedoo.fr/?editDatsId=994. The data for baro 1 and baro 2 are well present for the year 2012. The baro 2 data starts from 26/11/2012. In the table of appendix A, there is actually only one barometer in operation between September and November 2012 (baro 1). In contrast, there are two barometers in operation from December 2012 (baro 1 and baro 2). In the table of Appendix A, the periods are rounded per fortnight.

*http://mistrals.sedoo.fr/?editDatsId=993* : *SSC only available on request. Sensor name not specified in file. Long timespans contain of evidently non-operating sensor, which need to be masked as "nodata"*

Answer: For the response to the comment « *SSC only available on request* », see below the response to General issues b).

The sensor name is not specified in the file but it is specified in the metadata at http://mistrals.sedoo.fr/?editDatsId=993 and in Table 2. The model is the Visolid IQ 700 manufactured by WTW.

Overall there are few gaps in the time series. Currently (it is new since the first submission of the paper), the time series can be easily displayed in the BDOH database (https://bdoh.irstea.fr/OHM-CV/nos-donnees?site=Gazel+-+Cladu%C3%A8gne). The percentage of gaps per month is displayed in BDOH once a variable is selected (https://bdoh.irstea.fr/OHM-CV/CLA/TURB). There were a few more gaps in 2014 due to the more frequent burial of the turbidity sensor related to bedload during floods. During the procedure of quality control of the data, codes are used. These codes are defined as follows:
"v" for valid
"a" for absent (lack of information on the quality of the data)
"l" for data missing (due to a breakdown ...); the value of the data is then -9999
"I" for invalid (aberrant data that was deleted); the value of the data is then -9999
"d" for doubtful

This information was added in the README files located in the following directories:
…\zip1_auzon\zip1\surface water\hydrometric stations
…\zip1_auzon\zip1\surface water\stream sensors network

This README file was also added next to the data at http://mistrals.sedoo.fr/?editDatsId=993 and http://mistrals.sedoo.fr/?editDatsId=994

Flow and sediment transport are mainly concentrated during flood periods in the Mediterranean region (with high values of turbidity, typically in the range 1000 to 10000 ppm). We added the following information in the text (p.10 l.29): "Generally, for all inter-flood periods (marked by very low turbidity) the turbidity value was set to 0 during the procedure of data quality control even if there was a value different from 0 in the raw data. Indeed, the turbidimeters used in this study are designed for high turbidity range (typically 1000 to 10000 ppm) and are not capable of measuring accurately low turbidity (typically less than 50 ppm)."

*http://mistrals.sedoo.fr/?editDatsId=993 : not clear from what sensor this was derived. File gives a maximum discharge of 214 m³/s, which does not correspond to the hydrograph in Fig. 9.*

Answer: Thank you, well seen. Indeed, the figure 9 has been modified taking into account the last version of the stage-discharge rating curve. The information about the version of the rating curve is present next to the code of validity at each line of the data file. See for example:

20/04/2014 21:40:00;   0.223;v;courbe_03
20/04/2014 21:50:00;   0.223;v;courbe_03
20/04/2014 22:00:00;   0.223;v;courbe_03
20/04/2014 22:10:00;   0.235;v;courbe_03
20/04/2014 22:20:00;   0.235;v;courbe_03
20/04/2014 22:30:00;   0.223;v;courbe_03

*http://mistrals.sedoo.fr/?editDatsId=1349 : Table A: ends Oct 2014, website states Nov 2014*

Answer: yes, it is indicated 04/11/2014 at http://mistrals.sedoo.fr/?editDatsId=1349. However the periods are rounded per fortnight in the table of Appendix A. That is why there is this slight difference. This information was added to the legend of the table of Appendix A: "the periods are rounded per fortnight".

*http://mistrals.sedoo.fr/?editDatsId=1350 : theta: volumetric water content or relative saturation?*

Answer: Theta is volumetric water content (% Volume). This information was added in the README file located in the following directory:
…\zip1_auzon\zip1_auzon\soil\10 HS

This README file as well as a metadata file were also added next to the data at
http://mistrals.sedoo.fr/?editDatsId=1350

*General issues: a) information on timezones (UTC, CET, CEST?) should be provided wherever possible*

Answer: All the time series that are included in the bundled data "zip1_auzon.zip" and "zip2_auzon.zip" have been checked and are in UTC. We have completed the sentence in Section 5 (P.17, l.5-6) as follows: "The bundled data presents the advantage of gathering data in ASCII and cartesian format, in a single coordinate system and in the same timezone (UTC)." This information was also added on the landing page of the dataset: http://mistrals.sedoo.fr/?editDatsId=1438

*b) Providing "bundled data" seems to be a very useful idea. I could not abstract how to obtain these bundled data in the respective zip files, though.*

Answer: We do not understand the problem. A comment has been added to the interactive discussion on this subject on 17/11/2016 on the ESSD website (http://editor.copernicus.org/index.php/essd-2016-32-SC1.pdf?_mdl=msover_md&_jrl=386&_lcm=oc108lcm109w&_acm=get_comm_file&_ms=53835&c=115114&salt=1538399893683659499). The procedure for accessing the bundled data (see above the response to comment 9. for more details) had been explained in section 5 of the paper. In addition a "cover letter" submitted at the time of the initial submission of the paper indicated the procedure to download all the data (in particular the username and password of the account that enable to download the file "zip2_auzon.zip"). Once the "zip1_auzon.zip" and "zip2_auzon.zip" files are downloaded, a software such as 7zip is used to extract the compressed files.

*c) turbidity data is more useful when SSC-samples are provided*

Answer: OK, this is true that the SSC data were not present in the dataset at http://mistrals.sedoo.fr/?editDatsId=993 and in the file "zip1_auzon.zip". Ideally, we would have put the SSC time series instead of the turbidity time series. However, the relationship between turbidity in SSC is not unequivocal in small catchments such as the Gazel and Claduègne catchments. This relationship varies according to the size of the suspended sediment collected in the samples, which in turn varies according to the active erosion sources within the catchment. The task of transforming the turbidity time series into SSC time series is still in progress. Alternatively, we propose to provide the list of the samples taken by the automatic samplers with the corresponding SSCs measured in the laboratory.

Two files (GAZEL_SSC values for samples_2011-2014.xls and CLADUEGNE_SSC values for samples_2011-2014.xls) were added in the following directories (one file per station):
...\zip1_auzon\zip1\surface water\hydrometric stations\Gazel
...\zip1_auzon\zip1\surface water\hydrometric stations\Claduegne

These 2 files were also added at http://mistrals.sedoo.fr/?editDatsId=993

*d) Some data mentioned in the manuscript are apparently not part of the publication (e.g. geology map). It should be made clear wherever this is the case.*

Answer: In terms of in situ data, Table 5 lists all the individual datasets available, distinguishing between those that are under public status and those that are under the "associated scientists" status of the HyMeX data policy.

In terms of GIS descriptors, Table 3 lists all the existing data for the region of interest, according to our state of knowledge, distinguishing between those that are under public status and those that are "marketed products" commercialized by public or private organizations (such as IGN, BRGM, INRA). Of course we have these latter layers but we are not allowed to distribute them freely. However, we consider it is useful to inform readers of the existence of these data. We believe that the Editors should give their opinion whether or not these "marketed data" can be cited or not in the paper.

In contrast, Table 6 lists only the GIS descriptors under public status (ie those that are part of bundled data "zip1_auzon.zip"). This table provides information on accessibility to the individual datasets.

In Table 3, "subject to licensing terms" was replaced by "marketed product" as well as a note (*) in the foot page to indicate that these data are "not released in this study".

Additionally, the following sentence was completed (p.16 l.23) as follows: "All the individual datasets under the public status and the "associated status" of the HyMeX data policy presented in this study are listed in Table 5 and Table 6."

*e) The provided information on publication policy is misleading. From the entries I tested, only one required no registration and was truly public. Conversely, many datasets labeled as "Associated scientist" were completely inaccessible for me.*

Answer: see above the response to comment 9.

Annotated version of the paper

*p.1 l.31: always add protected whitespace between number and unit.*

Answer: ok, corrected

*p.2 l.12: consider replacing "opportunistic" ("concommittant"?)*

Answer: ok, replaced

*p.2 l.32: This and the following sentence are somewhat disconnected to the rest of the paragraph dealing with distributed models. Please relocate and improve line of argumentation.*

Answer: See above comment 3 as we have completely re-written the introduction.

*p.3 l.21: Please clarify or rephrase;   p.3l.24: essential for what?*

Answer: This paragraph was removed from the introduction as we considered it was too specific to rainfall processes. The introduction was refocused on the added value of the coupling between hydrometeorology and hydrology.

*p.3 l.28: Please please define what is "high" in this context*

Answer: We added "(~ km², ~ min)" after "High space-time resolution" p.1 l.27 and p.3 l.28

*p.3 l.30: adding the names of the datasets/catchments would illustrate this point*

Answer: ok, this is a good idea. We added the names of the datasets/catchments (p.3 l.28-33). It now reads as follows:

"The already published datasets consist of first order catchments ("Tarrawarra data set" (southeastern Australia): Western and Grayson, 1998), catchments where the observation period is exceptionally long ("Reynolds Creek Experimental Watershed" (northwestern USA): Slaughter et al., 2001; "Walnut Gulch Experimental Watershed" (southwestern USA): Renard et al., 2008; Stone et al., 2008; "Goodwater Creek Experimental Watershed and Salt River Basin" (midwestern USA): Baffaut et al., 2013), or catchments located in snow-dominated mountain ("Reynolds Creek Experimental Watershed" (northwestern USA): Reba et al., 2011; "Dry Creek Experimental Watershed" (northwestern USA): Kormos et al., 2014). In mesoscale catchments, such datasets are scarce ("Walnut Gulch Experimental Watershed" (southwestern USA): Goodrich et al., 1997; "Iowa River Basin" (north central USA): Gupta et al., 2010), especially in the Mediterranean region."

*p.4 l.1: "program" instead of "project"? Please specify the time span of the program. It should also be clear, if/which monitoring is still in operation.*

Answer: ok, we added the following information (p.4 l.3):

"A three-level nested experimental strategy was planned for the HyMeX program:

- A long-term observation period (LOP) lasting about 10 years (2010-2020) to gather and provide observations of the whole coupled system that support analysis of the seasonal-to-interannual variability of the water cycle through budget analyses.

- An enhanced observation period (EOP) lasting about 5 years (2011-2015), for both budget and process studies.

- Special observation periods (SOP) of several months, which aimed at providing detailed and specific observations to study key processes of the water cycle in specific Mediterranean regions, with emphases put on heavy precipitation systems and intense air-sea fluxes and dense water formation.

The FloodScale project (2012-2015) fits into the EOP and encompasses the SOP1 (Ducrocq et al., 2014) which took place from 5 September to 6 November 2012 and was dedicated to heavy precipitation and flash-floods."

*p.4 l.9: I cannot see why "the most extreme events" should be more difficult to observe in smaller catchments. "Extreme" events per definitionem are rare. This holds for all spatial scales.*

Answer: we decided to remove this sentence that was not very clear.

*p.4 l.16: I slightly disagree: high temporal resolution is also achieved by conventianal breakpoint data of raingauges*

Answer: we removed "and temporal". It now reads as follows:

"The observation strategy is reinforced by the deployment of conventional and polarimetric radars that provide precipitation measurements at spatial scales not properly resolved by rain gauges networks (Berne and Krajewski, 2013)."

*p.4 l.19: What does "uncertain" mean in this context? I agree that related measurements need to be made in high temporal resolution in small catchments - but this doesn't make the observations more uncertain. I also agree that stream gauging is challenging and prone to errors for higher stages - but this is independent of short response times. Please clarify.*

Answer: we substituted "an uncertain task" by "challenging". We meant that the observations are more uncertain in smaller catchment due to the very fast hydrological response. The variation of water level can be of 5 to 10 cm in less than 5 minutes whereas the time required for stream gauging using a surface velocity radar for example is typically 15 minutes. As a consequence, the variation of discharge during the stream gauging is far from negligible. Furthermore the presence of a bridge is rarely available in small catchments. Therefore there is more uncertainty in the surface velocity measurement and the cross-section is not as well sampled.

*p.4 l.25: "SOP", "SOP1" and "SOP-1" is used. If these are synonyms, unify; if not, explain. What is EOP?*

Answer: ok, see above the response to the comment "p.4 l.1".

*p.5 l.18: better: "over the plateau" or "in the higher areas"*

Answer: we substituted "over the relief" by "in the higher areas".

*p.8 l.14: This volumetric check would not detect a clogged funnel. I recommend plotting cumulated sums of rainfall of close stations against each other. This is how I spotted several unmarked instrument failures in your data.*

Answer: We thank the reviewer for providing this interesting advice. The volumetric check is actually done routinely. Moreover we provide an interactive plotting software on the OHMCV website to do so (http://www.ohmcv.fr/read-plot-tools.zip). Two explanatory notes on the validation procedure of the rainfall data as well as the plotting software (in R) used for this task have been added to the following directory in the bundled data:

…\zip1_auzon\zip1\rainfall\rain gauge Hpiconet\0_Documentation

Such documents are also available at http://mistrals.sedoo.fr/?editDatsId=656

Furthermore, this is now explicitly mentioned in the text as follows:

"A first step of the data quality control consists of comparing the total rain amount recorded by the tipping bucket-hobo system with those collected at the outlet of the rain gauge with a plastic tank of 30 l. The comparison of these two measurements shows that the relative difference remains below 10% for all the rain gauges and lower than 5% for most of them. Calibration is performed if an error of more than 5% is detected. In a second step, the temporal evolution of cumulated sums of rainfall

of close stations are plotted against each other. Clogged buckets never occurred thanks to the frequent maintenance (at least once a month)."

*p.8 l.19: What network?*

Answer: we completed the description as follows:

"The measured variables at each station differ depending on the network to which each station belongs. There are four types of networks for Météo-France stations present in this study: the "aeronautical synoptic" network (including LANAS-SYN), the RADOME-RESOME network with automatic stations also known as Automated Regional Network (including BERZEME-RAD), the network of automatic stations surveyed in real time (including ALBA-SA et ESCRINET) and the network of automatic stations interrogated in delayed time (including AUBENAS-SA et MIRABEL-SA). Each observing network has its own purposes so the needed variables are different. For instance, LANAS-SYN is the only station that provides surface pressure."

*p.8 l.22: better "which is installed 1.5 m above ground".*

Answer: ok, replaced

*p.9 l.5: better: "Geo-data"?*

Answer: We consider that the term "geo-data" can introduce an ambiguity between geography and geology. We had looked at how similar articles published in ESSD were organized and had decided to select the term "spatial characterization data". Therefore, we prefer to retain this term.

*p.10 l.12: "natural control points" = "natural cross sections"?*

Answer: "natural control points" was replaced by "natural controls" as it is formulated in World Meteorological Organisation. 2010. Manual on stream gauging. Volume I – Fieldwork, Volume II – Computation of Discharge. WMO-No. 1044, Geneva.

*p.10 l.31: Please specify how (decantation, filtration, filter pore size, ...)*

Answer: The procedure is explained in the paper by Navratil et al. (2011). Therefore a reference to this article was added as follows:

"The suspended sediment concentration (SSC), which represents the mass of solid divided by the volume of liquid (expressed in g/l), is estimated by measuring the volume of liquid present initially in the sample and weighting the solid fraction after drying it at 105 °C. The detailed procedure was given by Navratil et al. (2011) and resulted in a median relative uncertainty of 15%."

The following reference was also added in the section "References":

Navratil, O., Esteves, M., Legout, C., Gratiot, N., Némery, J., Willmore, S., Grangeon, T.: Global uncertainty analysis of suspended sediment monitoring using turbidimeter in a small mountainous river catchment. Journal of Hydrology, 398, 246-259, 2011.

*p.11 l.30: drying the water samples? Please clarify. Please also specify if SSC is given as m_sed/m_water or m_sed/(m_water+m_sed). Both*

Answer: The text was modified as follows:

"SSC were estimated following the procedure given above in sub-section "Hydrometric stations"."

*p.12 l8-10:* Answer: this lines were removed as proposed.

*p.12 l.20: better "assess the response to rainfall fields"*

Answer: ok, the sentence was modified as proposed. It now reads as follows:

"Given that rainfall is highly variable in space and time in this region, the observation system enables to assess the hydrological response to rainfall fields."

*p.13 l.18: better: "good areal representativity"?*

Answer: ok, "good accuracy on the average" was replaced by "good areal representativity".

*p.13 l.28: FDR, not TDR. Please add if/which if the internal default conversion tables (U -> theta) were used or custom calibration was performed.*

Answer: we thank referee 1 for this comment. "TDR" was replaced by "FDR". We added information on the conversion method used. It now reads as follows:

"Point volumetric soil moisture measurements were done using a portable three-prong (6 cm rod length) ML2X ThetaProbe sensor (Delta-T Devices Ltd, Cambridge, UK), which employs the frequency domain reflectometry (FDR) technique and the internal default conversion tables (to convert output voltage to volumetric soil moisture content) were used."

*p.13 l.32: I doubt that. How did you quantify this?*

Answer: we completed the sentence as follows:

"The accuracy given by the manufacturer is ± 0.01 $cm^3$ $cm^{-3}$."

*p.14 l.13: shorten: FDR-technology. Is the range of the instrument or the range of the data (the latter wouldn't quite fit here). How did you assess the accuracy?*

Answer: we thank referee 1 for this comment. We modified this paragraph. It now reads as follows:

"The selected sensors are capacitive probes (Decagon 10 HS) which employs the FDR technique and the internal default conversion tables (to convert output voltage to volumetric soil moisture content) were used. The range of volumetric soil moisture content of the instrument is between 0 and 0.57 $cm^3 cm^{-3}$ and the accuracy is ± 0.03 $cm^3 cm^{-3}$ according to the manufacturer."

*p.14 l.24: While these examples are certainly interesting, it seems there a numerous other studies based on the dataset. Thus, instead of picking examples, an overview table listing the key questions addressed and the corresponding publications would be more useful.*

Answer: see the response to comment 5 above.

*p.15 l.2: "small" is ambiguous in this context. Better use "high" or "low" in conjunction with resolution.*

Answer: we agree with this note. We modified the text (l.2 to 6) as follows:

"For all the scenarios, the results show that the error of the QPE increases with higher spatio-temporal resolutions. For the technique of kriging with external drift (merging radar and rain gauge data), there is a significant reduction in QPE error compared to the technique of ordinary kriging (using only rain gauge data) and this reduction is still more sensitive at higher spatio-temporal resolutions. Taking into account the data of the research rain gauge network (dense rainfall data) results in a reduction in QPE error. This reduction is similar to the decrease when adding the radar data, however the spatial structure of the errors and the rainfall estimates of these scenarios show large differences."

*p.16 l.10: "sedigraph" seems to be the more common expression.*

Answer: "sedimentograph" and "sedigraph" are both encountered in the literature. We finally decided to use "sediment flux at the outlet" for more clarity.

*p.16 l.18: This paragraph and the related figure does not really add any necessary information. I recommend removing it. If you still prefer to depict an example, consider selecting the Auzon station (as representing the entire catchment) and a finer temporal resolution to get an impression of the detailedness of the data.*

Answer: We do not totally agree with this comment. We recognize that this figure is not central to the paper. It is, moreover, little commented upon in the text of the paper. That is why we propose to put it in an appendix (→ Appendix B in the new version). In contrast, we think it is important to keep it for the following reasons:

- it provides an overview of the entire period 2011-2014 for 3 variables characteristic of the coupling "rainfall-discharge-suspended sediment" (i.e. the rain measured by an operational rain gauge located at the heart of the observation system, the discharge and the turbidity at the outlet of the catchment most densely monitored for hydrology). It is thus possible to identify easily within the period the dry or wet years, the most intense events...

- it illustrates the non-linear response between "rainfall-discharge-suspended sediment". A well-known example is that of 28/07/2013 which gave the largest rainfall amount in 6 hours and which is marked by a low hydrologic response but high turbidity.

The temporal resolution chosen for the rainfall (6h accumulated rainfall) is relevant for catchment basin because it corresponds approximately to the time of concentration of this catchment.

It is also important to note that we added the information "no data" on the time series of discharge (fig. b) and turbidity (fig. c) for the period from 01/01/2011 to 15/10/2011.

Ideally, we would have chosen to present the data of the station Auzon but there were two disadvantages:

- the measurement period begins in June 2013 for this station

- there is no measurement of suspended sediment for this station

Figure 8 shows the rainfall-discharge data obtained at the Auzon station for the event that produced the most important hydrological response of the period 2011-2014, even until to date.

*p.16 l.22: How can you tell this? If I understood correctly, monitoring only started in 2011, so computing return intervals from 5 years of data seems quite sketchy.*

Answer: Indeed, we have not yet done any statistical analysis to calculate the return periods of the different discharge values because the measurement period is too short. However, we did a work of gathering information among the inhabitants of the region. In particular, the person who lives at 50 m from the Claduègne station took photographs of the most important floods since 1990. The only other flood that produced an overflow in the floodplain occurred in October 1995 (see the photo below as well as that taken on 11/04/2014). This information was corroborated by interviews with farmers and residents who live by the river. So our estimate is fairly cautious. On the other hand, the measures have continued to date and we have not observed a larger flood since the end of 2014.

Note also that we added the following information to the text (p.16 l.22):

"(this return period was roughly estimated from archive photos and interviews with residents and farmers who live and work near the river)"

[Figure]

[Figure]

*p.17 l.2: I cannot confirm this: From the entries I tested, only one required no registration. Conversely, many datasets labelld as "Associated scientist" were completely inaccessible for me.*

Answer: see above the response to comment 9 for more details.

*p.17 l.3: Very useful idea. I could not abstract how to obtains these bundled data, though.*

Answer: see above the response to comment 9 for more details.

*p.24 l.30: The correct title of this paper is slightly different.*

Answer: We did the verification and did not see the problem.

**Table 1**

- *add instrument name*

Answer: This information was added in section 3.1.1 (p.7 l.4-5) rather than in Table 1 not to overload Table 1. It now reads as follows:

"The region of interest is covered by two operational S-band radars (Fig. 1): a conventional radar (Thomson MTO 2000S) located in Bollène (about 40 km away) and a polarimetric radar (Selex Meteor 600S) located in Nîmes (about 90 km away)."

- *add instrument name. Fig 3 lists 11 raingauges, not 21. Please clarify.*

Answer: we added the instrument name (Précis mécanique R3039 1000 cm²). We confirm that there are 21 rain gauges. Indeed, as it is said p.8 l.9-10, "At some locations, several rain gauges are clustered with interdistances of 10 up to 500 m as seen in the inset maps of Figure 4" in order to "sample rainfall at spatial scales ranging from tens of meters to tens of kilometres and at temporal scales ranging from 1 min to 1 day" (see p.8 l.8-9).

- *Fig. 3b shows only one radiation measurement, not two. Please clarify.*

Answer: we thank referee 1 for this comment. The number of pyranometers was set to 1 in Table 1. The sentence (p.8 l32-33) was also modified. It now reads as follows: "One out of the six considered stations is equipped with a pyranometer (CM11) providing global solar radiation values."

**Table 2**

- *remove "point"*

Answer: ok, it was done and Table 2 was completed to add the location of each instrument.

- *These data (and others) are not part of the publication. This should be made clear.*

Answer: ok, it was "subjected to licensing term" was replaced by "marketed product" and a note at the foot of the Table indicates that the "data are not released in this study". Of course we have these layers but we are not allowed to distribute them freely. However, we consider it is useful to inform readers of the existence of these data.

**Table 5**

- *this entry fits better into next category*

Answer: We decided to keep this field where it is since there is actually a measurement of atmospheric pressure in addition to the measurement of hydrostatic pressure. It is said p. 12 l.22: "The CTD-Divers and Mini-Divers measure the total pressure as they are not compensated (cableless instruments). An independent measurement of atmospheric pressure is therefore necessary for accurate barometric compensation".

**Figure 1**

- *This figure should concentrate on the Auzon-catchment, remove Gardon and zoom in. Verbose figure caption can be condensed into legend. Dashed circles need explanation. Inset map should be labelled "France" for clarity.*

Answer: We would like to keep this figure because it shows the geographical extension of the Cévennes - Vivarais Mediterranean Hydrometeorological Observatory (OHMCV) and the location of the different catchments studied in the Cévennes-Vivarais region during the Enhanced Observation Period of the HyMeX program. We consider important to have a regional vision and see all the nested scales until the Mediterranean Sea before zooming in on the study area (Figures 3, 4, 5 and 6).

Dashed circles were explained in the legend: "The two S-band operational radars with the range circles at 50 and 100 km (dashed circles) are also shown"

Inset map was labelled "France".

We prefer not to overload the figure with legends to keep clarity on the figure.

**Figure 3**

- *JPG-compression artefacts, use vector graphics if possible. Most fonts are too small to be read. Orange star (bz1) in c) is not explained in legend. Symbol for MRR in legend differs from the one in the map. 3 a) could be deleted, when Fig. 4 is modified.*

Answer: The figures will be produced in vector format for the final version of the paper if it is accepted (in .eps for example). At this point the figures were saved in .png and embedded in the paper but the quality was degraded during the production phase of the pdf file. We believe that this problem will be resolved in the future. We have worked hard on this figure, including on the size of the legends and it does not seem possible to enlarge the legend without losing information. Bz1 is actually represented by a green star but it looks like an orange star because of the low quality of the image. The MRR symbol seems different, but it is simply because there are several superimposed symbols. We prefer to keep the figures a, b, and c as they correspond to the general structure of the paper: a) precipitation data, b) other meteorological data and c) hydrological and sediment data. Figure 4 shows the multi-scale deployment of rain gauges, so the information is complementary. See also the response to the 2nd comment about Table 1.

**Figure 4**

- *If the central map would be replaced against 3a), the former could be omitted. Here, the background orthophoto is obsolete (if, for any reason, you want to keep it, give reference and add catchment boundaries).*

Answer: We have just explained in the response to the comment about Figure 3 why we wanted to keep both Figure 3 a) and Figure 4. We added the source of the orthophotos presented in the background in the legend of Figure 4 and we added the contours of the catchments on the map and in the legend. It now reads as follows:

"The boundaries of the Gazel and Claduègne catchments are displayed in black. Orthophotos are displayed in the background (source: IGN)."

**Figure 6**

*I did not read the related publication. Just based on the summary of this example study, it is hard to understand:*

*\* "final error standard deviation" is not a well defined term. Assuming that this relates to Kriging variance, the images of c) and d) are misleading, as one could conclude that the error of KED is higher (especially within the detected cell) and thus the method is worse. This needs commenting.*

*\* OK as the simpler method should be "a)" in the first row, not "b)"*

*I suggest removing the example or improving the explanations.*

Answer: We thank the reviewer for this important remark. We propose:

- to replace "final error standard deviations by "error standard deviations" in the figure caption.

- to add this sentence in the text just after the mention to Fig 6: "Figure 6.a. and 6.b. illustrate the added-value of radar data to capture the spatial structure of rainfall even if, as for this case study (Fig. 6.c and 6.d.), QPE's error standard deviation may be sometimes larger than for ordinary kriging. Boudevillain et al. (2016) showed that, in general, QPE's errors are significantly reduced with the technique of kriging with external drift."

Figure 6 was modified as proposed: OK as the simpler method was moved to "a)" in the first row and KED was moved to "b" in the second row. The caption was modified consequently.

**Appendix A**

- *very useful table. Please try using landscape page layout to avoid splitting the table. Multiple mismatches with actual data vailability, see test_access.xlsx*

Answer: We thank referee 1 for this comment. We have tried to use landscape page layout in the new version of the paper. We added the following note at the foot of the page:

"For each year from 2011 to 2014, the number of operating instruments is indicated by fortnight period (the months are numbered from 1 to 12)."

We guess that this reason explain most of the discrepancies noted in the file test_access.xlsx

We also added the following information in the Table and at the foot of the page:

" * Means that the instrument(s) is(are) still running at the date of publication of the paper and at least for the period of LOP (long-term observation period, 2010-2020) of the HyMeX program"

- *It seems, the table contains the number of operating instruments. Instead, the number of valid time series (presumably less than b) would be more useful (e.g. 20 instead of 21 for hpiconet in 2014)*

Answer: Thank you for this comment, it is indeed the number of valid chronicles that are displayed in this Table. However, this was not the case for the "Hpiconet rain gauges". Instead it was indicated the number of instruments deployed in situ. Therefore we have updated this line in the Table with the number of valid chronicles.

---

## Author Comment (AC2) · 16 Dec 2016

**Final author comments to referee 1 comments on "A high space-time resolution dataset linking meteorological forcing and hydro-sedimentary response in a mesoscale Mediterranean catchment (Auzon) of the Ardèche region, France" by Nord et al.**

In the following, the reviewer comments appear in black italic and our answers are provided in blue. When there are quotations from the text of the article, they appear in quotation marks and the new or corrected parts are highlighted in yellow.

essd-2016-32-RC2.pdf

*Another thing that needs to be discussed is the policy for data access and data re-use. Concerning access, I already expressed some concerns. More specifically, it needs to be made clear how datasets are \*accessible\* and under what conditions the data can be \*reused\*: access: \* truly public (ie. directly accessible) \* after registration \* with additional costs (i.e. cannot be part of this publication) reuse: \* free (i.e. Public Domain, Creative Commons or similar) \* additional restrictions Specifically, I would like to discuss the implications of the HYMEX data policy C1(http://mistrals.sedoo.fr/HyMeX/Data-Policy/HyMeX_DataPolicy.pdf), which seems to govern the largest part of the data.*

Answer: See above the response to comment 9 (in the response to essd-2016-32-RC1.pdf).

As stated above, the data under public status are not subject to the HyMeX data policy which applies only to data under "associated scientists" or "core users" status. The data under public status represent the majority of the data associated with this paper. Most of these data are bundled in the "zip1_auzon.zip" file and are directly accessible to a ftp without any step of identification.

On the other hand, all the individual datasets under public status are accessible independently via the links indicated in Tables 5 and 6. It is true that there is an authentication step for the individual datasets stored In the HyMeX database but the access is immediate and free after the creation of an account. The step of creation of this account has been modified to comply with basic rules of ethics.

For individual datasets under the "associated scientists" status of the HyMeX data policy, the rules for accessing and using these data (*http://mistrals.sedoo.fr/HyMeX/Data-Policy/HyMeX_DataPolicy.pdf*) must be accepted by the future user of the data during the step of creation of an account in the database HyMeX (http://mistrals.sedoo.fr/User-Account-Creation/). For more transparency, the evaluation process of the new account is now explained in the web page of the form for the account creation. The data remain free but are subject to constraints explained in the data policy. These constraints are common for data of this type which are usually marketed by Météo France.

*a) Section 3 §4 states that data can be made public domain. However, imposing additional obligations (section 4) contradicts to the common notion of PD as being completely free. b) Section 4 §2 restricts the use of the data to objectives of the HYMEX programme. c) While the correct attribution of the data being used of course is scientific best practice (section 4, §5), the license also*

*requires to offer collaboration to the PI (§6) and co-authorship (§7). d) Derived data and publications need to be provided to HYMEX (§11-13). In my opinion,*

*a) needs clarification.*

Answer: We think we have made clarifications with the answer to comment 9 (in the response to essd-2016-32-RC1.pdf) and the additional answer just above.

*b) likely restricts the use for some scientific research questions and should be removed.*

Answer: This constraint applies only to individual datasets under the "associated scientists" status of the HyMeX data policy (9 of 41 individual datasets in this study), the other individual datasets under public status are not subject to it (32 of 41 in this study). We are aware that this constraint limits the scope of the individual datasets under the "associated scientists" status. We have ensured that a maximum number of individual datasets are under the public status. However some datasets remain under this status despite our efforts. It will not be possible to amend the data policy.

*c) with §6 and §7 is ethically undue.  The authors of the data have already claimed their academic rewards by (then) having published the ESSD-paper and it being cited in case of re-use. While any researcher may be happy to contact the PI and invite him to collaborate, it cannot be mandatory.*

Answer: We recognize this may be a problem, we have done our utmost to ensure that it covers a limited number of individual datasets (only the individual datasets under the "associated status" of the HyMeX data policy). However, we consider that the individual datasets concerned are important to integrate into the study.

*d) loosely corresponds to a Creative Commons Attribution-NonCommercial 4.0 International (CC BY-NC 4.0), which is acceptable.*

Answer: Again, this constraint applies only to individual datasets under the "associated scientists" status of the HyMeX data policy (9 of 41 in this study), the other individual datasets under public status are not subject to it (32 of 41 in this study).

*These are, of course, ethical decisions and thus open to debate. The requirements of ESSD are quite diffuse about that (data needs to be "openly accessible (cost-free for readers)"). I specifically ask the other reviewers and the editor to express their opinion on these points.*

Answer:  We had already exchanged a lot on this subject with the editors before submitting the paper. We have tried to take into account their remarks as well as those of the reviewers as best we could.

---

## Author Comment (AC3) · 16 Dec 2016

**Final author comments to referee 2 comments on "A high space-time resolution dataset linking meteorological forcing and hydro-sedimentary response in a mesoscale Mediterranean catchment (Auzon) of the Ardèche region, France" by Nord et al.**

In the following, the reviewer comments appear in black italic and our answers are provided in blue. When there are quotations from the text of the article, they appear in quotation marks and the new or corrected parts are highlighted in yellow.

essd-2016-32-RC3.pdf

*This work describes a comprehensive dataset that includes hydrometeorological, and hydrological data collected from a number of sensors (in situ and remote) over a mesoscale catchment in Mediterranean. The number of observed variables, the various sensors involved and the space/time resolution of this comprehensive dataset makes it a unique contribution to research community. I agree with the authors that such a dataset can serve as an excellent benchmark for evaluating and improving process based models used for understanding triggering rainfall properties, runoff generation mechanisms and erosion processes during Mediterranean floods. My recommendation is to accept the paper for publication in ESSD journal.*

Answer: We thank referee 2 for this positive appraisal of the dataset presented in this paper.

*Below I list only some minor points/corrections that may help to improve information provided in some specific parts of the text. However, I want to share also the concerns raised by reviewer#1 regarding the conditions that may potentially apply on the use of this dataset. My understanding from reading the manuscript is that the "non public" data require simply a registration to HyMeX before they can be retrieved and used. If there is more into this (see points/concerns of reviewer#1) it needs to be clarified and dataset needs to be subsequently adjusted to include only data that are unconditionally available.*

Answer: The HyMeX database requires creating an account even for a user who wants to download only data under public status. We recognize that this step certainly constitutes a complication for a fully open and prompt access to the data. However, it also makes possible to trace the users of the data and contact them in case of an important updating of the datasets. The main problem was that this creation of account required the user to validate the HyMeX data policy whereas the data under public status are not subject to this policy (indeed the HyMeX data policy only applies to data under "associated scientists" or "core users" status). This was an error and was not ethically founded. As a result, with the help of the managers of the HyMeX database, we have modified the step of creation of account by this way (http://mistrals.sedoo.fr/User-Account-Creation/): users wanting to download only datasets under public status no longer need to validate the HyMeX data policy. A simple registration (name, affiliation, e-mail) is necessary to create an account (an e-mail with their login and password is instantaneously sent to their e-mail address) and then access directly to public data.

[Figure]

If users want to use datasets under "associated scientists" status, two situations are possible:

1) Either they have registered previously for access to public data only. In this case, they must go to their account (http://mistrals.sedoo.fr/Your-Account/) and complete the HyMeX registration application.

2) Or they register for the 1st time. They must go to http://mistrals.sedoo.fr/User-Account-Creation/ and do not check the box corresponding to "Simple registration (name, affiliation, country) for direct access to public data only."

In both cases, they will arrive on a page where they have to describe the planned work in more than 350 characters and they have to validate the HyMeX data policy by accepting various statements. A sentence has been added at the foot of the page to explain the procedure of validation: "Registrations are handled by the MISTRALS Executive Comittee following the criteria established by the HyMeX International Scientific Steering Committee. Registration is reviewed within 7 days and is valid for 3 years." The database managers have calculated that, since the creation of the HyMeX database, the average period of validation for applications is: 4.43 days.

Most of the individual datasets (32 of 41) have "public" access. A few of them (9 of 41) are under the "Associated scientists" status of the HyMeX data policy.

Nevertheless, to facilitate the use of the data and to avoid downloading each individual datasets (41 in total), a bundling service was provided. The bundled data presents the advantage of gathering data in ASCII and cartesian format, in a single coordinate system, and in the same timezone (UTC). The bundled data were selected for the spatial and temporal windows presented in the paper since some individual datasets have different extents. It represented an important task. As explained in section 5 Data availability, the bundled data were organized in two zip files: the "zip1_auzon.zip" and "zip2_auzon.zip" files available from the link to the data repository included in the abstract of the paper (http://mistrals.sedoo.fr/MISTRALS/?editDatsId=1438).

Access to the file "zip1_auzon.zip" is totally public (see the flag "Direct access to public data" in the page http://mistrals.sedoo.fr/MISTRALS/?editDatsId=1438) and does not require any identification. The data can be downloaded directly without any step of identification. In contrast, access to the file "zip2_auzon.zip" is under "associated scientists" status and is subject to the HyMeX data policy. We created a special account to allow editors and reviewers to download this file "zip2_auzon.zip". We had sent a "cover letter" at the time of first submission of the paper to explain the procedure to download all the bundled data (in particular the username and password of the account that enable to download the file "zip2_auzon.zip ").

*Minor points*

*1. P1L29: ". . .quantity and type of sensors", consider improve wording*

Answer: We recognize that the understanding of the text was not straightforward. We have rephrased some of the abstract based on the comments of the two referees to improve the clarity. It now reads as follows:

"A comprehensive hydrometeorological dataset is presented spanning the period 1 January 2011-31 December 2014 to improve the understanding of the hydrological processes leading to flash floods and the relation between rainfall, runoff, erosion and sediment transport in a mesoscale catchment (Auzon, 116 km²) of the Mediterranean region. Badlands are present in the Auzon catchment and well connected to high gradient channels of bedrock rivers which promotes the transfer of suspended solids downstream. The number of observed variables, the various sensors involved (both in situ and remote) and the space-time resolution (~ km², ~ min) of this comprehensive dataset makes it a unique contribution to research communities focused on hydrometeorology, surface hydrology and erosion. Given that rainfall is highly variable in space and time in this region, the observation system enables to assess the hydrological response to rainfall fields. Indeed (i) rainfall data are provided by rain gauges (both a research network of 21 rain gauges with 5 min time step and an operational network of 10 rain gauges with 5 min or 1 h time step), S-band Doppler dual-polarization radars (1 km², 5 min resolution), disdrometers (16 sensors working at 30 s or 1 min time step) and Micro Rain Radars (5 sensors, 100 m height resolution). Additionally, during the special observation period (SOP-1) of the HyMeX (Hydrological Cycle in the Mediterranean Experiment) project, two X-band radars provided precipitation measurements at very fine spatial and temporal scales (1 ha, 5 min). (ii) Other meteorological data are taken from the operational surface weather observation stations of Météo-France (including 2-m air temperature, atmospheric pressure, 2-m relative humidity, 10-m wind speed and direction, global radiation) at the hourly time resolution (6 stations in the region of interest). (iii) The monitoring of surface hydrology and suspended sediment is multi-scale and based on nested catchments. Three hydrometric stations estimate water discharge at a 2 to 10 min time resolution. Two of these stations also measure additional physico-chemical variables (turbidity, temperature, conductivity) and water samples are collected automatically during floods allowing further geochemical characterization of water and suspended solids. Two experimental plots monitor overland flow and erosion at 1 min time resolution on a hillslope with vineyard. A network of 11 sensors installed in the intermittent hydrographic network continuously measures water level and water temperature in headwater subcatchments (from 0.17 km² to 116 km²) at a time resolution of 2-5 min. A network of soil moisture sensors enable the continuous measurement of soil volumetric water content at 20 min time resolution at 9 sites. Additionally, concomitant observations (soil moisture measurements and stream gauging) were performed during floods between 2012 and 2014. Finally, this dataset is considered appropriate for understanding the rainfall variability in time and space at fine scales, improving areal rainfall estimations and progressing in distributed hydrological and erosion modelling."

*2. P2L4: ". . .measure water discharge. . .", do you mean water stage or "estimate water discharge"?*

Answer: ok, "measure" was replaced by "estimate" (see above)

*3. P2L26: "Indeed, the water and sediment discharges simulated by distributed models. . .are generally poorer. . .". It is stated as a universal truth. I suggest to revise statement to mention that "many studies have shown. . .", for example.*

Answer: ok, this was corrected. We have also completely re-written and shortened the introduction as suggested by referee 1. It now reads as follows:

"The Mediterranean area is prone to intense rainfall events, sometimes triggering flash floods that may have dramatic consequences (Ruin et al., 2008). Flash floods are the consequence of short, high-intensity rainfalls mainly of spatially confined convective origin and often enhanced by orography (Borga et al., 2014). As such, flash floods usually impact basins less than 1000 km² (Marchi et al., 2010). In medium-scale Mediterranean catchments, the control exerted by the amount of rainfall and its intensity and variability on the generation of runoff and the erosional processes operating at different scales is of major importance (Navratil et al., 2012; Marra et al., 2014; Tuset et al., 2015). Assisting stakeholders in implementing efficient soil conservation and river management measures implies understanding the processes and the factors that control surface runoff, develop modelling approaches able to provide reliable flow separations, localize sediment sources and sinks, and predict the space-time dynamics of sediment and associated contaminant within the catchment. This requires taking into account the space-time variability of rainfall events, using spatially distributed models coupling hydrology and mass transfers.

Although the interest of distributed models is recognized for understanding the inner behaviour of the catchment (i.e. pathways and transit times), many studies have shown that their reliability do not meet the expectations. Indeed, the water and sediment discharges simulated by distributed models at the outlet of the catchment are generally poorer than the results simulated by lumped models (Jetten et al., 2003; Reed et al., 2004; de Vente el al., 2013). To date there are various difficulties that hinder the potential of distributed models (e.g. Cea et al., 2016) such as the large number of parameters, the definition of some parameters which are difficult to measure, the high non-linearity of the equations, the interaction between input parameters, the uncertainty in the experimental measurements and input data, the space-time variability of the physical processes, and the lack of comprehensive field data available for initialization and calibration. Thus the deployment of multi-scale observation systems over a period of several years in medium catchments and the release of the collected datasets as open data with metadata on how the data have been collected, quality assured, and their associated uncertainties (Weiler and Beven, 2015) is of crucial importance to address the current limitations of distributed models.

High space-time resolution (~ km², ~ min) datasets linking meteorological forcing and hydro-sedimentary response are rare in scientific literature because of the high number and diversity of types of sensors required for measuring rainfall and surface hydrology. The already published datasets consist of first order catchments ("Tarrawarra data set" (southeastern Australia): Western and Grayson, 1998), catchments where the observation period is exceptionally long ("Reynolds Creek Experimental Watershed" (northwestern USA): Slaughter et al., 2001; "Walnut Gulch Experimental Watershed" (southwestern USA): Renard et al., 2008; Stone et al., 2008; "Goodwater Creek Experimental Watershed and Salt River Basin" (midwestern USA): Baffaut et al., 2013), or catchments located in snow-dominated mountain ("Reynolds Creek Experimental Watershed" (northwestern USA): Reba et al., 2011; "Dry Creek Experimental Watershed" (northwestern USA): Kormos et al., 2014). In mesoscale catchments, such datasets are scarce ("Walnut Gulch Experimental Watershed" (southwestern USA): Goodrich et al., 1997; "Iowa River Basin" (north central USA): Gupta et al., 2010), especially in the Mediterranean region.

This study is part of the FloodScale project (Braud et al., 2014), which is a contribution to the HyMeX program (Hydrological Cycle in the Mediterranean Experiment, Drobinski et al., 2014), a 10-year multidisciplinary program on the Mediterranean water cycle. A three-level nested experimental strategy was planned for the HyMeX program:

- A long-term observation period (LOP) lasting about 10 years (2010-2020) to gather and provide observations of the whole coupled system that support analysis of the seasonal-to-interannual variability of the water cycle through budget analyses.

- An enhanced observation period (EOP) lasting about 5 years (2011-2015), for both budget and process studies.

- Special observation periods (SOP) of several months, which aimed at providing detailed and specific observations to study key processes of the water cycle in specific Mediterranean regions, with emphases put on heavy precipitation systems and intense air-sea fluxes and dense water formation.

The FloodScale project (2012-2015) fits into the EOP and encompasses the SOP1 (Ducrocq et al., 2014) which took place from 5 September to 6 November 2012 and was dedicated to heavy precipitation and flash-floods. This study focuses on nested scales that range from the hillslope to the medium catchment scale all belonging to the Cévennes - Vivarais Mediterranean Hydrometeorological Observatory (OHMCV) (Boudevillain et al., 2011). It is located in Ardèche, in a region with a high gradient in annual rainfall (e.g. Molinié et al., 2012). The observation system has been operated by different teams from various countries during the SOP1 and EOP: LTHE, IRSTEA Lyon, EPFL, Wageningen University, LAMP and Météo-France. The dataset includes precipitation and weather data, soil moisture data, runoff and soil erosion data, hydrologic and suspended sediment response data, surface water quality data, and GIS data.

The duration of the observations presented here (4 years, from 1 January 2011 to 31 December 2014) allows the characterization of the standard catchment behaviour and provides the opportunity to observe less ordinary events with processes that are specific to flash floods and to characterize possible threshold effects that are not observed in small to moderate events. The observation strategy is reinforced by the deployment of conventional and polarimetric radars that provide precipitation measurements at spatial scales not properly resolved by rain gauges networks (Berne and Krajewski, 2013). A special effort was dedicated to soil moisture measurements and stream gauging during floods. These opportunistic observations made possible by a real-time warning system enable to watch transient processes like runoff, to monitor the increase of water content in soil and to gauge high discharges in small to medium catchments, which is challenging due to the very short response times of such systems. This allows documenting the upper ends of stage-discharge rating curves that are generally extrapolated at high values.

The paper presents the acquired datasets to make them accessible to the scientific community and make their use easier and wider. The authors are convinced that the published datasets can serve as a benchmark for hydrological distributed modelling applied to the Mediterranean area. The paper is organized as follows: Section 2 presents the location of the studied catchment and its context (geology, climatology, land use, pedology). Section 3 describes the observation system (instruments and measured variables) and is organized in three subsections: i) hydrometeorological data, ii) spatial characterization data, iii) hydrological and sediment data. Finally, in Section 4, the first studies that provide preliminary answers to the scientific questions selected in the introduction are highlighted."

*4. P2L28: "this raises the question of the improvement. . .." Improve wording/syntax.*

Answer: This sentence was finally deleted (see in the previous point the new version of the introduction).

*5. P3L6: ". . .to exceed the current limitation. . ." consider changing this to "to improve distributed models" or "to address current limitations"*

Answer: ok, "exceed" was replaced by "address" (see above the new version of the introduction).

*6. P3L16: I am not sure about the "leaser extent on-board satellites. . .". What is the basis for stating that? Especially since you are referring to rainfall all around the world where in many parts satellites are the only source of information.*

Answer: This paragraph was removed from the introduction as it was considered too specific to rainfall processes by the authors. Our objective has been to shorten the introduction and gain in clarity. The introduction was refocused on the added value of the coupling between hydrometeorology and hydrology.

*7. P3L21: "rainfall is not steady. . ." what do you mean by stead rainfall?*

Answer: same answer as to comment 6.

*8. P3L22: "..the scales of operational rain gauge. . .". I am not sure what is your reference here but at a global level the typical temporal scale of rain gauge observations is daily.*

Answer: same answer as to comment 6.

*9. P3L24: "operational radars is essential", for what? The sentence ends rather abruptly.*

Answer: same answer as to comment 6.

*10. P4L25: Have you previously define what SOP and EOP stands for?*

Answer: We thank referee 2 for this note. Indeed, we had not defined precisely SOP and EOP previously in the text. Therefore we have described the 3 phases of the HyMeX program (with their respective periods of application), which makes it possible to better situate this study in relation to the HyMeX program (see above the new version of the introduction).

*11. P7L19:which one is the MXPol? Is it the EPFL-LTE? It is a bit confusing.*

Answer: We added the name of each radar directly after its first introduction in the text to gain in clarity. We also added the instrument name for the S-band radars. It now reads as follows:

"The region of interest is covered by two operational S-band radars (Fig. 1): a conventional radar (Thomson MTO 2000S) located in Bollène (about 40 km away) and a polarimetric radar (Selex Meteor 600S) located in Nîmes (about 90 km away). Their visibility over the Auzon catchment is however hindered by the topography and the lowest beam is at about 2 km above the ground. These operational radar, managed by Météo-France, provided data (radar reflectivity and rain rate estimates) over the entire period of interest. To complement these radars and monitor the small-scale variability of precipitation, two additional X-band research radars were deployed during HyMeX SOP1 (Fig. 3.a), providing measurements at a resolution of about 100x100 m². A fast-scanning radar (WR-10X+), managed by LaMP, provided rapid Plan Position Indicator (PPI) scans (every 3 minutes) at one elevation. EPFL-LTE managed a Mobile X-band Polarimetric (MXPol) radar, that provided a combination of Range Height Indicator (RHI) and PPI scans of polarimetric variables every 5 min. These two research radars enabled the monitoring of low level precipitation over the Auzon catchment. Their maximum range should vary between 30 and 40 km (the range represented in Fig. 3.a is only qualitative). Finally, 5 Micro Rain Radars (MRR), provided by CNRM, LaMP and OSUG, were deployed in combination during Fall 2012 and Fall 2013 at three locations in the region of interest to document the vertical profile of precipitation. These Doppler FW-CW vertical pointing radars measuring the Doppler spectra enable to study the vertical structure of rainfall as well as the associated microphysical processes in relation with the orography (Zwiebel et al., 2015). More detailed information about the operational and research radar systems involved in HyMeX can be found in Bousquet et al (2015). The operational radar processing algorithms are described in Tabary (2007), while the data from the MXPol radar are processed following the steps described in Schneebeli et al. (2014) and Griazioli et al. (2015). The characteristics of MXPol are given by Schneebeli et al (2013) and Mishra et al. (2016)."

12. P8L12:"..is made thanks to a Hobo. . ." consider revising and avoiding "thanks to"

Answer: ok, we corrected the sentence. It now reads as follows:

"Each rain gauge is connected to a Hobo or a Campbell data logger."

13. P14L27: Not clear what you mean by "radar-rain gauge" here. Do you refer to a merging technique?  OR just a geostatistical-based interpolation to obtain "radar-like" QPE from gauges. Please clarify.

Answer: We refer to a merging method of radar and rain gauge data through Kriging with External Drift. The sentence "Geostatistical techniques (ordinary kriging and kriging with external drift) were used to obtain radar–rain gauge quantitative precipitation estimates (QPE's)." was modified as follows: "Geostatistical techniques (raingauge ordinary kriging, as well as merging of radar and raingauge data through kriging with external drift) were used to obtain quantitative precipitation estimates (QPE's)."

14. P15L31: ". . .we saw during the events of the 2014", how did you see that? Please explain.

Answer: we recognize that the sentence was not clear. It was removed and replaced by the following sentence:"Figure 8 shows the hydrograph and the associated uncertainty for the different rating

curves presented in Figure 7 for the 04 November 2014 event, the largest flood recorded during the period 2011-2014."

*15. P16L12-13: NSE and PBIAS are not defined previously.*

Answer: ok, we completed the text as follows:

"The Nash – Sutcliffe efficiency (NSE), the percent bias (PBIAS)…"

*16. P16L22: "5-10 yr return period". This is interesting and I am wondering what is the basis for this estimate? Are there any flood frequency curves available for the area? It would be interesting to have this infor for other flood events in the record.*

Answer: Indeed, we have not yet done any statistical analysis to calculate the return periods of the different discharge values because the measurement period is too short. However, we did a work of gathering information among the inhabitants of the region. In particular, the person who lives at 50 m from the Claduègne station took photographs of the most important floods since 1990. The only other flood that produced an overflow in the floodplain occurred in October 1995 (see the photo below). This information was corroborated by interviews with farmers and residents who live by the river. So our estimate is fairly cautious. On the other hand, the measures have continued to date and we have not observed a larger flood since the end of 2014.

Note also that we added the following information to the text (l.16):

"(this return period was estimated from archive photos and interviews with residents and farmers who live and work near the river)"

[Figure]

[Figure]

04/11/2014 14:06

17. *P17L28: ". . .quality (size, fall velocity. . .", please revise because size and velocity of hydrometeors is not a qualitative measure.*

Answer: This sentence was finally deleted